# Predicting European cities' climate mitigation performance using machine learning

Angel Hsu [1,2,3] ✉, Xuewei Wang[1,2,3], Jonas Tan[4], Wayne Toh[4] & Nihit Goyal [5]

Although cities have risen to prominence as climate actors, emissions' data scarcity has been the primary challenge to evaluating their performance. Here we develop a scalable, replicable machine learning approach for evaluating the mitigation performance for nearly all local administrative areas in Europe from 2001-2018. By combining publicly available, spatially explicit environmental and socio-economic data with self-reported emissions data from European cities, we predict annual carbon dioxide emissions to explore trends in city-scale mitigation performance. We find that European cities participating in transnational climate initiatives have likely decreased emissions since 2001, with slightly more than half likely to have achieved their 2020 emissions reduction target. Cities who report emissions data are more likely to have achieved greater reductions than those who fail to report any data. Despite its limitations, our model provides a replicable, scalable starting point for understanding city-level climate emissions mitigation performance.

Cities have in recent years risen in prominence on the global sustainability policy agenda, as researchers and policy-makers have increasingly focused on urban jurisdictions as powerful policy actors in their own right. More than 10,000 of the world's cities are pledging various forms of climate mitigation, adaptation, and financing actions, and in many instances these municipalities participate in multiple voluntary transnational climate initiatives[1]. As part of these initiatives' requirements, in accordance with national government directives[2], or on their own volition, cities articulate strategies and policies to tackle climate change mitigation and, less frequently, adaptation. Cities predominantly put forth mitigation strategies centered on greenhouse gas emission reduction targets, often achieved through policies focused on increasing the use of sustainable transport, enhancing the efficiency of lighting in public and municipal buildings, adopting energy efficiency standards, promoting climate awareness to encourage citizen action, and other areas[3,4].

There are thousands of current strategies and policies detailing urban mitigation efforts, yet, as Milojevic-Dupont & Creutzig[5] point out, there is little understanding of these actions' effects. These knowledge gaps cause policymakers to be "disoriented on which measures are

adequate and impactful" in urban areas and uncertain which "everyday decisions" regarding planning or infrastructure investments should be made to achieve mitigation targets. Little is known about the emission reductions from common urban climate policies and strategies, a missing block of vital information acknowledged in Chapter 12 on Human Settlements in the Fifth Assessment Report (AR5) of the Intergovernmental Panel on Climate Change (IPCC)[5,6].

Scholars have argued that cities' involvement in transnational climate governance "can accelerate their actions to curb GHG emissions under certain conditions"[7]. The evidence in support of this claim is scarce, making it hard to predict precisely what conditions would have this effect. Transnational climate initiatives typically require reporting of climate action plans and regular monitoring in the form of emissions inventories to assess whether mitigation goals are met, yet in practice only a small fraction of subnational actors meet these requirements[8,9]. Hsu et al[10] found that out of more than 9000 cities that were signatories to the EU Covenant of Mayors for Climate and Energy (EUCoM) initiative, only ~15% had reported any emissions data, and even fewer (around 11%) had reported both a baseline emissions

[1]Department of Public Policy, University of North Carolina at Chapel Hill, Chapel Hill, USA. [2]Data-Driven EnviroLab, University of North Carolina at Chapel Hill, Chapel Hill, USA. [3]Institute for the Environment, University of North Carolina at Chapel Hill, Chapel Hill, USA. [4]Yale-NUS College, Singapore, Singapore. [5]Faculty of Technology, Policy, and Management, Delft University of Technology, Delft, Netherlands. ✉e-mail: angel.hsu@unc.edu

inventory and an additional year of inventory emissions data needed to track progress towards voluntary reduction targets. When emissions data are available, they are frequently incomparable due to the limited availability of datapoints, a general lack of transparency regarding underlying methodologies, and the lack of standardized accounting approaches. Ibrahim et al.[11] evaluated seven distinct city-scale greenhouse gas emissions inventory protocols and methodologies and concluded that a common reporting standard or approach is needed for cities. Differences in the various standards' definitions—e.g., for emission scopes, particularly in Scope 3 supply chain emissions—must be addressed so that participants emissions' data can be appropriately compared.

Recent advances in machine learning (ML), a general class of non-parametric, non-linear statistical modeling approaches and computational algorithms usually applied to large-scale datasets to simulate human learning, could help us overcome these tricky emissions data challenges[12]. In this study, we employ a ML-driven approach to estimating and evaluating the mitigation performance of nearly all local and municipal actors in the European Union and the United Kingdom from 2001 to 2018. Our method develops a process for identifying spatial boundaries and geospatial predictors for each local and municipal government participating in the EUCoM, one of the largest voluntary transnational climate governance initiatives, and then utilizing the self-reported carbon emissions inventory data from ~6000 EUCoM cities as training data in an extreme gradient boosting model. To our knowledge, our resulting dataset is the most comprehensive time series dataset used to evaluate city-level carbon emissions and mitigation performance. We apply these data to evaluate the performance of three groups of European cities: "reporting" cities that have reported at least one year of emissions data; "participating" cities that have pledged voluntary climate action but have not reported any emissions data; and last, "external" cities representing local administrative units (LAUs) that are not participants.

## Results
### City-level predictors of climate emissions
Figure 1a shows the correlation between the city-level dependent (i.e., self-reported "emissions") and independent variables (i.e., heating degree days, fossil-fuel $CO_2$, GDP per capita, etc.). We found a strong positive correlation between reported emissions inventory data and stationary fossil-fuel $CO_2$ emissions from the Open-Data Inventory for Anthropogenic Carbon dioxide (ODIAC)[13] ($r^2 = 0.81$), as well as between emissions and population ($r^2 = 0.89$). Population and stationary fossil-fuel $CO_2$ emissions were also highly correlated ($r^2 = 0.79$), confirming prior studies that demonstrate through the use of nighttime lights intensity the relationships between these data and energy consumption, economic activity, and fossil-fuel emissions[14]. Our analysis did not show strong relationships between self-reported emissions data and GDP per capita ($r^2 = 0.03$) or with fine particulate air pollution (PM2.5; $r^2 = 0$). We determined that stationary fossil-fuel $CO_2$ emissions and population were the primary predictors of cities' self-reported emissions data with the highest contribution or importance to our emissions model (Fig. 1b). Figure 1b shows the gain value of the importance of each of the top six features we considered. The gain values are determined by the amount each attribute split improves the model's performance, weighted by the number of observations for the node. See Methods for more description about the grid search process and parameter tuning to determine the final model.

We predicted emissions for around 92,636 cities or local administrative units (LAUs) where we had underlying spatial data (Supplementary Table 2). Figure 2 presents scatterplots of cities' self-reported emissions data compared to our model's predicted emissions data. The resulting $r^2 = 0.91$ indicates our model is strongly predictive overall of cities' self-reported emissions inventories. We further validated our predicted emissions with other studies that report emissions data for European cities, including Moran et al.[15], who estimate 2018 direct (Scope 1) emissions for more than 100,000 European cities and Nangini et al.[16], who combine self-reported inventories with other data for 343 global cities. We found fair correlation ($r^2 = 0.57$ with Moran et al.[15]; $r^2 = 0.62$ with Nangini et al.[16]) between our predicted data and these other studies (Supplementary Fig. 7). Figure 2b also shows the self-reported emissions data vs. predicted emissions data by country, which allows closer examination of potential eccentricities in our model or the predicted data. For some countries, such as Ukraine, our model performs less well ($r^2 = 0.02$), and for some particular cities, the predicted emissions are higher than what the cities themselves have

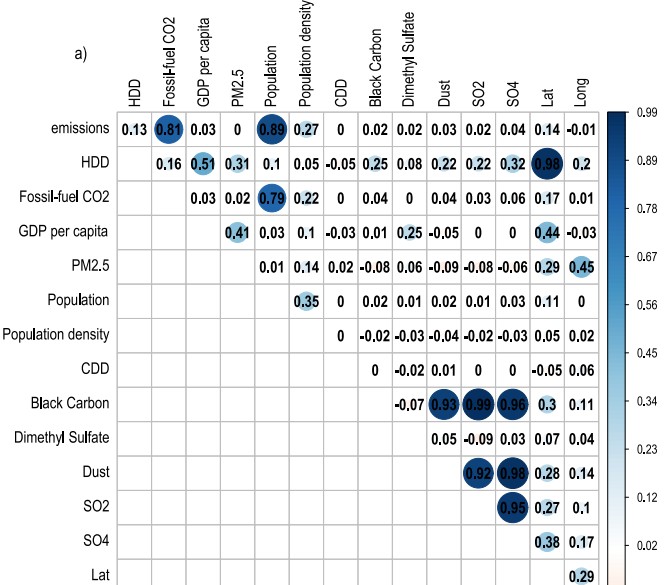

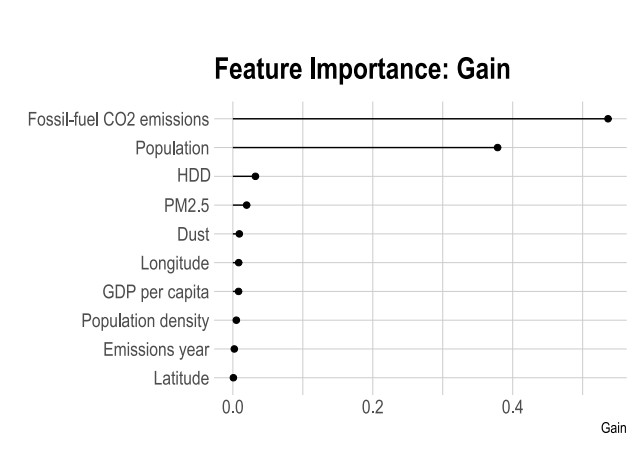

**Fig. 1 | Predictors of European cities' emissions. a** Correlation matrices showing the relationship between various predictors of urban climate emissions. **b** Importance of various predictor variables to the emissions' prediction model.

The more an attribute is utilized to make decisions in the XGBoost model, the higher its feature importance is determined.

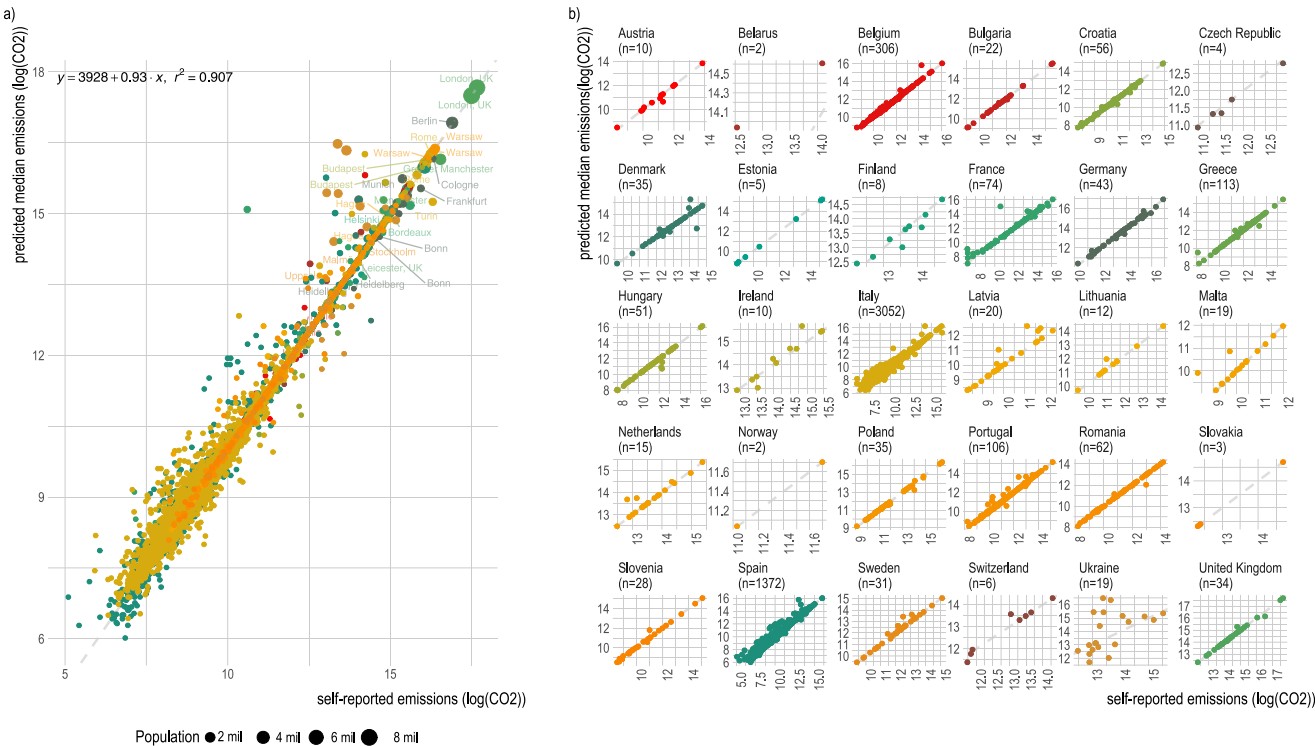

**Fig. 2 | Predicted emissions compared to self-reported emissions.** Scatterplot of self-reported emissions ($n = 6961$ self-reported emissions data-points from cities reporting to the EUCoM used in the model training) compared to the predicted median emissions for each actor from the model on a log scale. **a** All self-reported emissions inventories (in log tons $CO_2$) of all actors versus the predicted emissions data (in log tons $CO_2$); **b** Country-by-country facets of self-reported vs. predicted emissions where there were more than 1 datapoint. The number of cities listed in the country panels slightly vary from Supplementary Table 3 since Supplementary Table 3 includes both cities reporting emissions data and those that do not.

reported. For instance, our model predicts annual emissions nearly three times higher than Lyon's self-reported emissions. Further inspection of one of these outliers, Lyon, a city of 445,000 people in France, reports an emissions inventory of around 22,000 tons, translating in per capita emissions of <0.05 tons, far below the national average of 5.4 tons per person[17].

**Trends in performance**

Utilizing available time series data of underlying predictors, we generated likely annual emissions ranges for all cities and local administrative units where data were available from 2001 to 2018. Illustrating the output of our model, Fig. 3 shows time series for three selected cities of varying population sizes: Waimes in Belgium (population: 8932), Tolosa in Spain (population: 17,575), and London in the United Kingdom (population: 8.9 million). These data were then analyzed for trends in annual per capita emissions reduction over the time period from 2001 to 2018 for cities participating in the EUCoM that report emissions data (reporting cities), those that do not report (participating cities), and for all LAUs in Europe (external cities).

Overall, we find that EUCoM cities have on average, likely reduced annual per capita emissions from 2001 to 2018 ($-0.96 \pm 1.88\%$) and from 2005 to 2018 ($-0.53 \pm 3.3\%$), compared to external cities that on average, are likely to have not experienced much change in emissions ($0.18 \pm 2.5\%$ from 2001 to 2018 and $0.18 \pm 3.2$ from 2005 to 2018; Table 1). While 74% of EUCoM cities are likely to have reduced emissions, only 53% of external cities are likely to have experienced a negative trend in emissions reductions. We interpret these emission trend differences between EUCoM cities and external LAUs with caution, however, noting the differences most

notably in population between EUCoM ($32,720 \pm 181,348$ inhabitants for reporting cities; $35,318 \pm 171$ for participating cities) and external LAUs, which tend to be on average much smaller ($4433 \pm 16,870$ inhabitants) (Supplementary Table 2; Supplementary Fig. 6). Descriptive statistics (Supplementary Table 2) and distributions (Supplementary Fig. 6) describing the three groups of cities in our analysis illustrate that EUCoM cities tend to have more sizeable stationary fossil-fuel carbon dioxide emissions and be larger in population and population density than external cities, which could explain differences in their emissions trends, since larger cities with higher levels of GDP per capita have been shown to have more ambitious climate plans[2,18].

Within the EUCoM cities, we find that cities self-reporting emissions inventory data (75% of EUCoM cities) are likely to have achieved greater average emissions reductions compared to participating cities that have not reported a baseline or monitoring emissions inventory ($-1.3 \pm 1.7$ vs. $0.2 \pm 1.9$ annual per capita emissions reductions between 2001 and 2018; Table 1). These results suggest that participating EUCoM cities are likely to have achieved the same mitigation performance as external cities. EUCoM cities that have pledged relatively more ambitious mitigation targets, exceeding the EU's 2020 mitigation target of 20% reduction from 1990 levels, are likely to have achieved greater annualized per capita emissions reductions compared to cities with a relatively less ambitious mitigation target ($-1.4 \pm 1.7$ vs. $-0.6 \pm .20$ from 2001 to 2018; Table 1). Finally, EUCoM cities likely on track (e.g., sufficiently reducing emissions in line with required emissions to reach their declared 2020 emissions reduction target, see Methods for further details) to achieve their 2020 emission reduction targets (52% of EUCoM cities) likely have

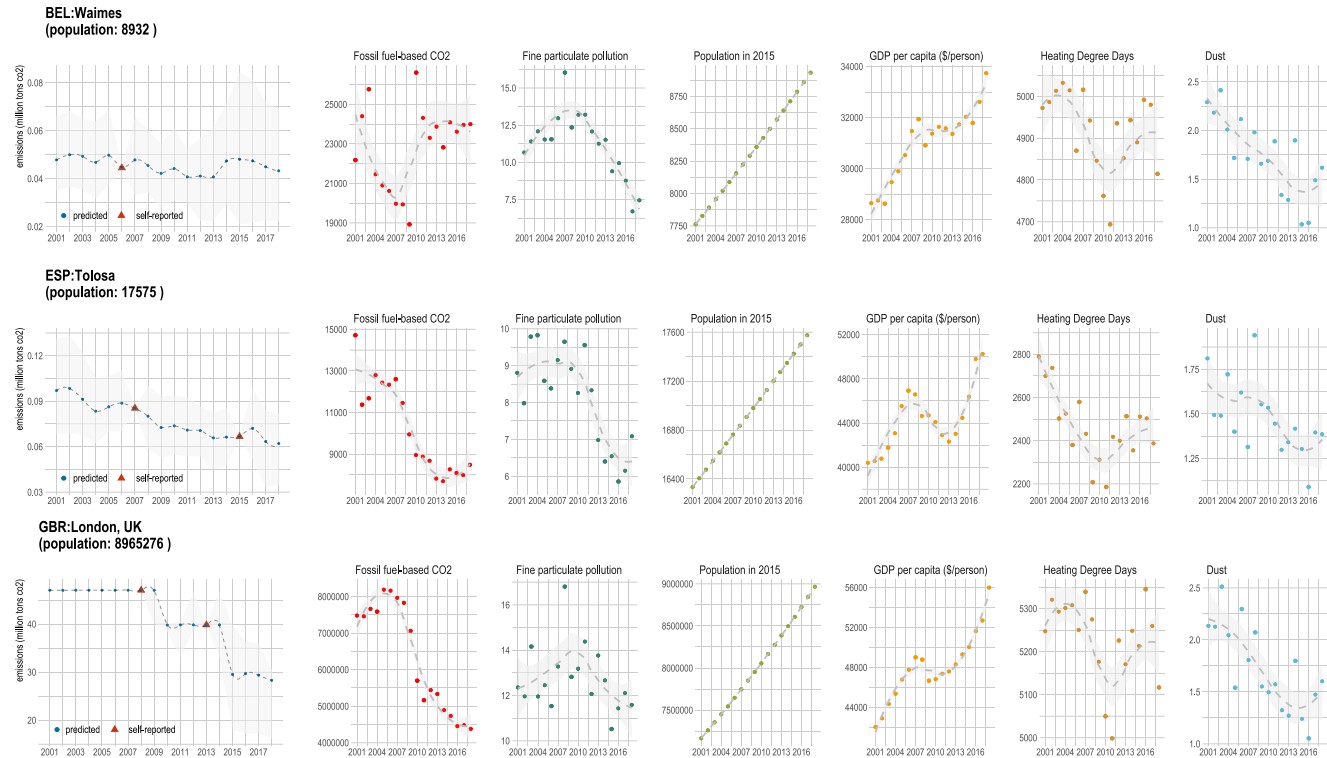

**Fig. 3 | Predicted, self-reported emissions, and primary predictor variables for three cities of varying population sizes.** Waimes in Belgium, Tolosa in Spain, and London in the United Kingdom were selected to represent cities of varying population sizes.

experienced the greatest emission reductions (−1.8 ± 2.5 vs. 0.5 ± 1.6 from 2001 to 2018; Table 1). EUCoM cities not on track (48% of EUCoM cities) have likely experienced a slight growth in annual per capita emissions.

While we lack sufficient controls and data to causally isolate whether participation in the EUCoM led to these emissions mitigation trends, an interrupted time series (ITS) analysis, which models whether a policy intervention or program may have resulted in a measurable change in an outcome variable after its implementation[19,20], can shed some light on whether EUCoM cities' emissions reductions primarily occurred after they joined the initiative, accounting for differences in population density, GDP per capita, etc. (see Methods for further

## Table 1 | Difference in annual per capita emissions reduction trend between different comparison groups

| | Mean ± sd trend (1) | Mean ± sd trend (2) | Mean difference | Standard error |
|---|---|---|---|---|
| **(1) EUCoM cities vs. (2) External Cities** | | | | |
| 2001–2018 trend | −0.96 ± 1.88 | 0.18 ± 2.5 | 1.14*** | 0.0003 |
| 2005–2018 trend | −0.5 ± 3.3 | 0.18 ± 3.2 | 0.68*** | 0.0004 |
| **(1) Reporting EUCoM cities vs. (2) participating EUCoM cities** | | | | |
| 2001–2018 trend | −1.3 ± 1.7 | 0.2 ± 1.9 | 1.55*** | 0.001 |
| 2005–2018 trend | −0.75 ± 3.5 | 0.3 ± 2.5 | 1.03*** | 0.002 |
| **(1) Ambitious EUCoM cities versus (2) unambitious EUCoM cities** | | | | |
| 2001–2018 trend | −1.4 ± 1.7 | −0.6 ± 2.0 | 0.87*** | 0.001 |
| 2005–2018 trend | −0.82 ± 3.5 | −0.3 ± 3.0 | 0.52*** | 0.002 |
| **(1) On-track EUCoM cities versus (2) Not on-track EUCoM cities** | | | | |
| 2001–2018 trend | −1.8 ± 2.5 | 0.5 ± 1.6 | 2.72*** | 0.0004 |
| 2005–2018 trend | −2.2 ± 0.9 | 0.9 ± 3.5 | 2.66*** | 0.0007 |

Difference in means were statistically compared using a two-sided *t*-test.
EUCoM cities include all reporting and participating cities unless otherwise noted.
***$p < 0.01$.

details). We find that each year following a cities' adhesion to the EUCoM initiative is associated with a slight −0.164 (standard error, or se: 0.039) annual percentage change in per capita emissions (Fig. 4). Confirming our comparison between city groups (Table 1), the ITS regression further demonstrates the significance of an emissions inventory ($p < 0.01$), where reporting cities have likely achieved a −1.24 (se: 0.396) annual percentage change in per capita emissions (Table 2). The level of the 2020 emissions reduction target, although slightly significant ($p < 0.05$), does not seem to have much of an additional effect on annual percentage change in per capita emissions (Table 2).

We observe differences in performance by country. Figures 5 and 6 compare the performance of participating EUCoM cities versus all other LAUs by country. In some countries, EUCoM cities, such as those in Sweden and Denmark, on average have had higher annual per capita reduction trends than external city counterparts. In others, such as the Netherlands and the United Kingdom, EUCoM cities appear to be underperforming compared to other cities (Fig. 4), as evidenced by comparing the distributions of annual per capita emissions reductions for both groups of cities. This result may reflect the fact that the national governments of Denmark and the United Kingdom require local climate action plans from municipalities[2], suggesting that external cities in these countries may be reducing emissions to meet national regulations and requirements. Italy and Spain, where most of the EUCoM cities are located, appear to have relatively comparable performance for both groups (Italy = 64%; Spain = 50%; Supplementary Table 3). Scandinavian countries lead in terms of countries with the highest proportion of cities on track (80% in Denmark; 53% in Finland and 70% in Norway). Spain also boasts a large proportion of cities on track, with 68%. Countries where cities perform similarly are closer to the diagonal line in Supplementary Fig. 8, suggesting that the mean annual per capita emissions reduction trends are similar among EUCoM and external cities. Countries above the diagonal are those

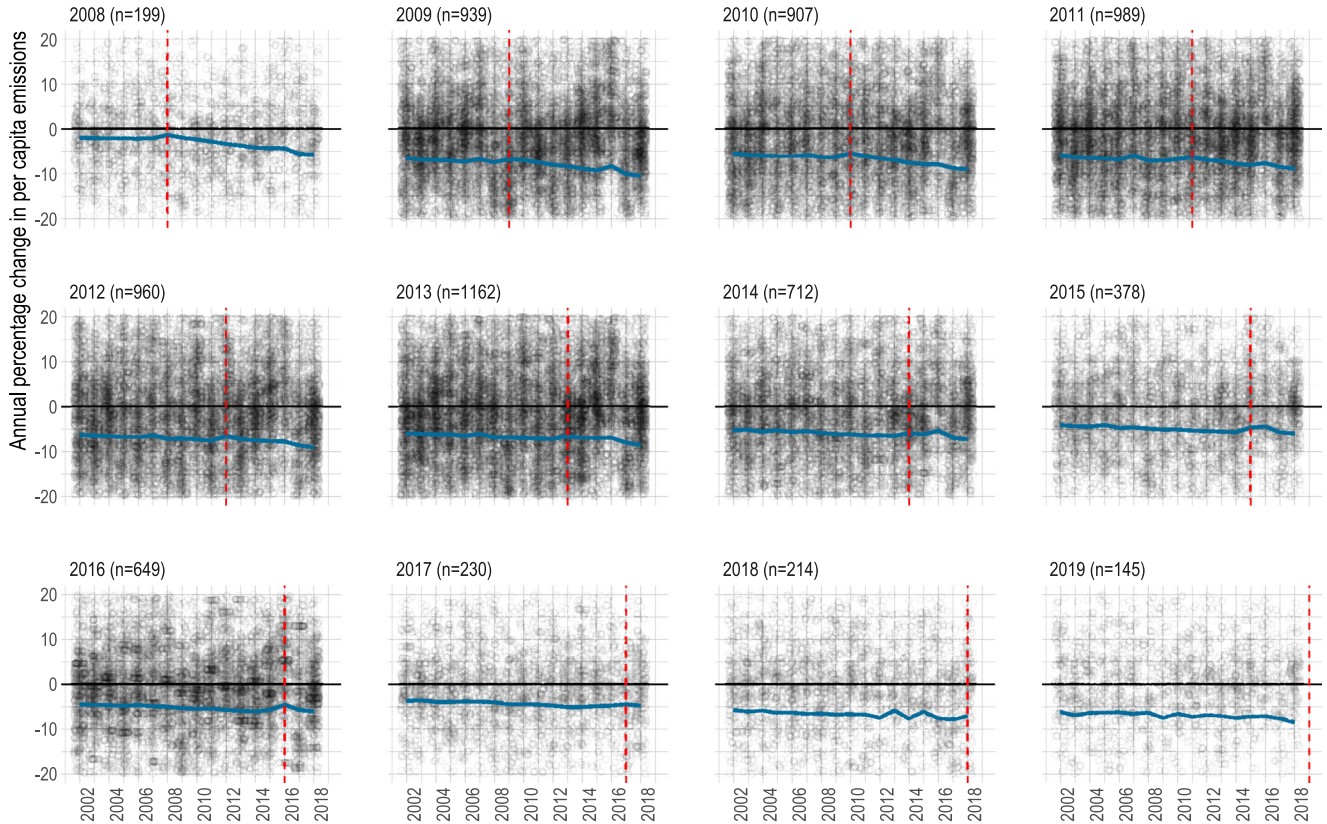

**Fig. 4 | Effect of participating in the EU Covenant of Mayors for Climate and Energy.** Annual percentage per capita change in emissions for EUCoM cities (plotted points) with predicted annual percentage per capita change in emissions determined by interrupted time series analysis (blue line). Panels include data for cities that joined the EUCoM in that specific year only, indicated by the red vertical lines.

where EUCoM cities have achieved greater annual per capita emissions reductions than their non-EUCoM counterparts and include countries like Finland, Slovakia, France, Germany, Italy among others.

## Discussion

Despite a measurable increase in urban climate governance scholarship over the past decade, gaps in understanding outcomes for transnational climate initiatives have persisted, particularly for smaller cities and on a systematic basis[21]. Part of this gap is due to data availability and comparability, which limit researchers' ability to trace causal impacts or linkages between the processes and institutions of transnational urban climate governance initiatives to outcomes[21,22]. To

**Table 2 | Results of interrupted time series analysis**

| | Dependent variable: annual percentage change |
|---|---|
| Time | 1.416*** (0.342) |
| Joined | 0.286 (0.191) |
| Time Since Joining EUCoM | −0.164*** (0.039) |
| log(Population density) | 1.395*** (0.032) |
| log(GDP per capita) | 2.432*** (0.198) |
| Predicted emissions per capita | 0.076*** (0.007) |
| 2020 reduction target | −0.015** (0.006) |
| Mitigation plan | −0.046 (0.400) |
| Mitigation inventory | −1.239*** (0.396) |
| n | 102,787 |

Standard errors are in parentheses. The regressions include country and year fixed effects.
***p < 0.01.

address this shortcoming, this study has developed a machine learning (ML)-based framework to predict more than 90,000 European cities' emissions on an annual basis from 2001 to 2018 to examine likely mitigation performance trends. By utilizing globally gridded, spatially explicit predictor variables that are measured consistently and regularly and available self-reported emissions inventories, our ML-based model is able to explain 90% of the variation ($r^2 = 0.90$) between self-reported emissions inventory data from recording EUCoM cities and predicted emissions values, validated through comparisons with other studies that have produced city-level carbon emission estimates for a single year. Not without its limitations (see Limitations), our model provides a replicable, scalable starting point for understanding city-level climate emissions mitigation performance. It also provides a method of evaluating and validating cities' self-reported emissions. Since some cities may erroneously report inventories or choose to selectively report emissions sources, our approach can help to spot outliers or potential reporting issues.

From our model's predicted emissions data, we examined annual per capita emissions trends that revealed insights that warrant further exploration. First, of the roughly 8000 European cities that participate in one of the largest voluntary transnational climate initiatives—the EU Covenant of Mayors for Climate and Energy (EUCoM)—most (74%) are likely to have reduced emissions from 2001 to 2018, with slightly more than half likely to achieved to have achieved their 2020 emission reduction target. Cities that self-report emissions data are likely to have reduced more than cities that have not reported emissions data, a finding that could be due to the fact that, as Rivas et al.[23] found, EUCoM municipalities that monitor emissions tend to also have started implementing plans earlier and are typically "frontrunners" with more climate action experience[23]. Cities with more ambitious mitigation targets and those on track to achieving their 2020 mitigation targets

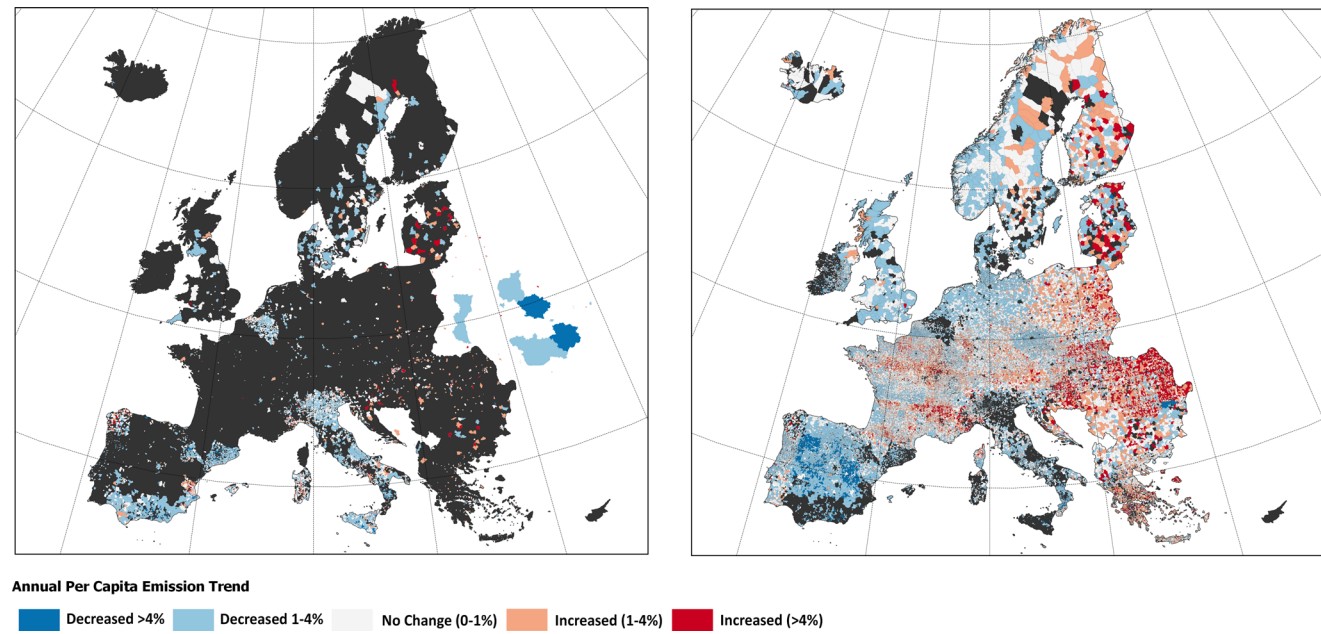

**Annual Per Capita Emission Trend**

| ■ Decreased >4% | ■ Decreased 1-4% | No Change (0-1%) | ■ Increased (1-4%) | ■ Increased (>4%) |

**Fig. 5 | Maps of European cities' emissions trends.** Annual per capita emissions reduction trend from 2001 to 2018 for cities participating in the EUCoM (left) and all other external cities (right).

have likely achieved the greatest annual per capita emissions reductions compared to their counterparts. Our findings here echo results of previous studies of EUCoM cities[10,23,24]. Kona et al.[24], for example, analyzed 315 reporting cities and found that they had reduced emissions by 23% on average, while our results are comparable (-1% annualized per capita emissions from 2001 to 2018). In our 2020 study of 1066 EUCoM cities reporting at least two emission inventories, we found 60% on track to achieve their emission reduction targets, while this study found similar results with 52% of EUCoM cities likely to have achieved their 2020 mitigation targets. Rivas et al.[23] suggest that ambition and monitoring may be linked—municipalities that tend to be more ambitious in their targets tend to not have reported monitoring inventories, echoing Hsu et al.[10] in finding a disconnect between ambition and performance.

While our study does not speak to causal mechanisms of the predicted emission trends, nor whether there are endogenous conditions that may explain why EUCoM cities have experienced on average slightly greater annual per capita reductions than their external non-EUCoM counterparts, it does suggest some insights relevant for urban climate governance and transnational climate initiatives. First, since emissions inventories and monitoring protocols are considered hallmarks of effective local governments' climate mitigation plans[25], the ability to monitor and report emissions are likely indicators of capacity and achievement. We measured significant differences in annualized per capita emissions reductions between reporting cities and participating cities that fail to report any emissions data, which are likely to be more similar to external EU cities in their emissions trajectories compared to reporting cities. Second, while assessing emissions trends as an outcome variable does not provide a "measure of effort"[26] nor describe the myriad inputs and factors that have led to a particular outcome, monitoring and reporting emissions inventories indicates a "means of implementation"[26] for evaluating an entity's progress towards a policy outcome like climate mitigation. These findings regarding linkages between monitoring and performance have implications for driving improvements in subnational climate mitigation, suggesting investments in monitoring are one likely predictor of success. Rivas et al.[23] found that the odds of emissions monitoring are 2.24 times higher when a local authority provides financial support for a climate plan's implementation.

Data describing mitigation outcomes then allow for identification of "general conditions of successful implementation" and reverse engineering of causal pathways that led to the emissions reductions. Our dataset and replicable, scalable ML-framework can subsequently provide a first step towards disentangling which specific measures, or none at all, led to the observed emissions reductions. Since we were limited to data on cities' population, GDP, air pollution, and fossil-fuel $CO_2$ emissions, our analysis cannot account for other underlying structural differences (e.g., variation in governance institutions, etc.) that may further elucidate differences in emissions outcomes, since climate change action and policies are "deeply entwined with other policy agendas."[27,28] In addition, our model produces one of many possible emissions pathways cities may have experienced, based on the limited, available predictors we used.

## Future research
Since the availability of self-reported emissions inventory data at the subnational level is primarily constrained to Europe, future studies must broaden the search for relevant datasets and proxies that can fill this gap, particularly for capacity- and resource-constrained entities in the Global South[29–31]. Actors in these countries face limitations (e.g., expertise, lack of clearly designated roles in relevant government agencies for producing inventories, insufficient documentation and archival systems) and technical issues (e.g., incomplete or non-existent activity data or lack of experimental data for developing countries or technology-specific emission factors) for producing emissions inventories[10,32]. Our next step is to expand our approach to a set of subnational jurisdictions outside of Europe to produce a global dataset for cities participating in transnational climate initiatives, as recorded in Hsu et al.'s[1] dataset of more than 10,000 cities and regional governments. We find compelling evidence that large-scale, geospatial datasets can be applied to estimate city-level carbon dioxide emissions, even for small city actors that comprise the majority of participants in the EUCoM, although more data and an expanded scope can better stress test the applicability of the model beyond Europe. Our method bridges the gap between these globally available, remote-sensing derived geospatial datasets to city-scale actors, a shortcoming Pan et al.[33] note in fossil-fuel $CO_2$ datasets like the ODIAC inventory,

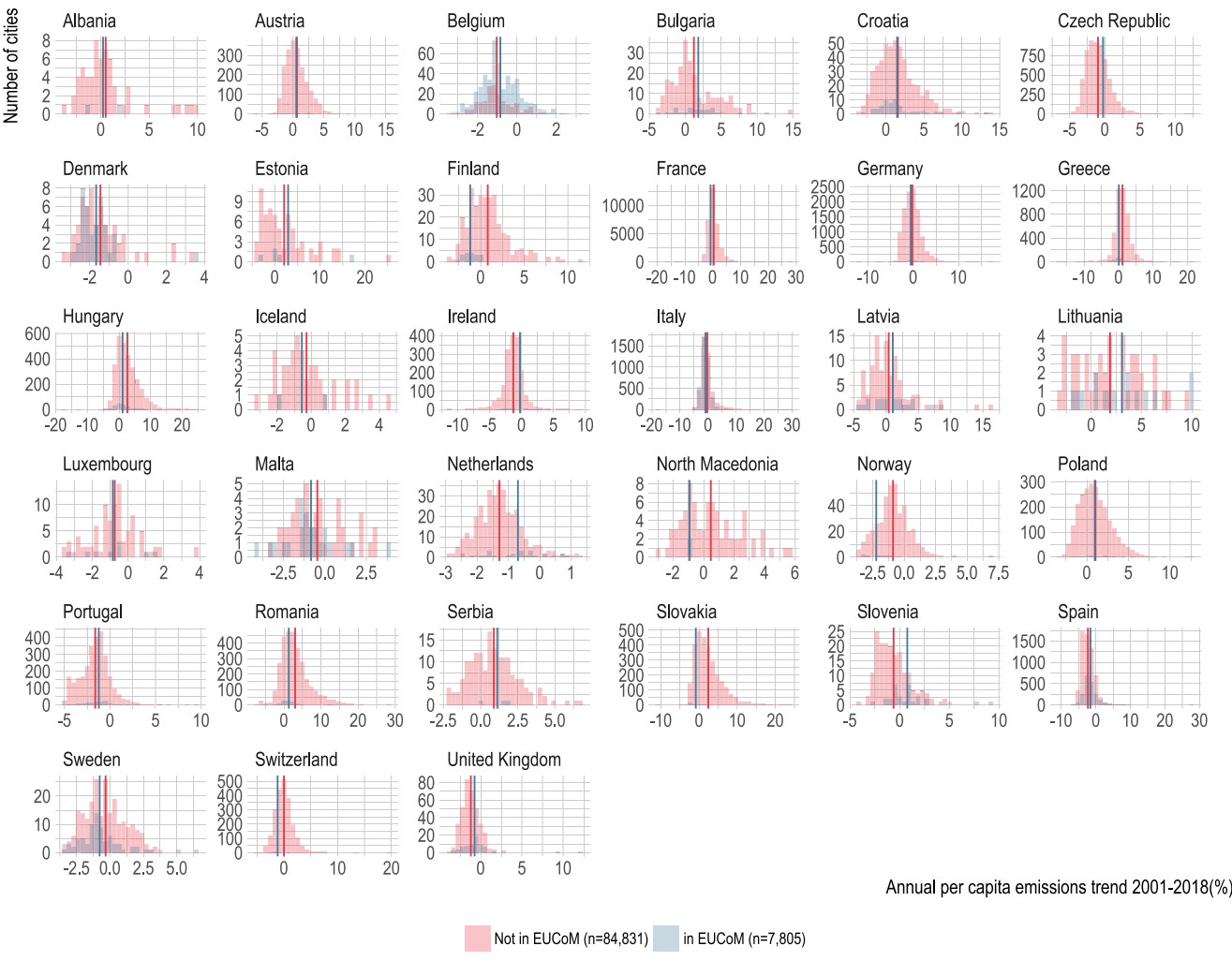

**Fig. 6 | Comparing emission reductions between city groups.** Distributions of annual per capita emissions reductions between cities in the EUCoM and external cities. Negative numbers indicate emissions reductions and mean annual per capita emissions trends for each group are designated with vertical lines in each panel.

which primarily distributes national fossil-fuel CO₂ emissions spatially based on satellite measurements of light-output intensity, and which may not correctly attribute emissions to subnational actors. Last, more research deeply evaluating the mitigation policies different groups of cities adopt to achieve emissions reductions and mitigation outcomes is required to inform urban planning and future climate policy development is needed.

## Limitations

This study is certainly not without its limitations. There are a few areas of uncertainty that could affect the validity of our predictions and results. First, we assume that the self-reported emissions inventories from the EUCoM actors are a valid source of data to train our model and predict others' emissions. We used the "verified" dataset of self-reported emissions data for 6,200 cities that had reported emissions inventory data evaluated by the European Commission's Joint Research Centre[34]. Although Kona et al.[34] applied a series of statistical checks to validate these reported emissions inventories, they note several limitations. Since the focus of the EUCoM is on greenhouse gas emissions related to sectors where a local authority has power to influence through sectoral and policy measures, participating cities only report emissions from selected sources (e.g., energy consumption for buildings, transport and local energy generation, industrial sources not already covered by the EU Emissions Trading Scheme, and waste/wastewater[35]). Kona et al.[34] acknowledge that the EUCoM

inventories were "never meant to be a method to create exhaustive inventories of all emission sources in the territory or to deal with emissions already included in national-scale control initiatives, such as the EU Emissions Trading System (ETS) mechanisms." Therefore a second limitation is that there are emissions sources and sectors that could be missing from EUCoM cities' inventories, particularly if a city doesn't have the capacity to measure those emissions or they deem certain emissions sources to not be of material importance for management purposes. Third, reporting cities' use of different emissions factors, estimation methodologies, and reporting boundaries add uncertainty to the use of their inventories as training data, and we found that some prediction "outliers" could be attributed to the fact that the initial self-reported emissions data could be the result of calculation or reporting error by the city itself[23]. Rivas et al.[23] note this limitation, particularly with regards to data sometimes reported with missing emissions factors, which then need to be filled with national or regional factors and could affect the accuracy of the final estimation[23]. Fourth, we assume that the spatial boundaries of EUCoM and external cities remain static over the time period, while these may have changed over time. If the boundaries have changed or are incorrectly identified or matched with a city, their predictions could be inaccurate. Fifth, while we observed significant differences between different cities' emissions, there may be some fundamental differences between these groups of cities that would account for mitigation performance that our model is unable to tease out (e.g., whether reporting EUCoM cities

are fundamentally different from non-reporting or non-participating cities in terms of geography, culture, or government that would drive their emissions trends).

Last, there are limitations to machine learning-based approaches for prediction, which have been identified and classified by Kapoor and Narayanan[36]. Since machine learning approaches are inherently stochastic[37], the introduction of randomness to enhance model generalizability, which is seen as an advantage of ML approaches compared to traditional gaussian regression methods, risks a model's potential reproducibility[36]. Our estimates, therefore only represent a likely bootstrapped median emissions level using the specific parameters tuned on the training subsample and set at a particular seed or initialization by the computing environment. In particular, our predictions of LAUs that report no emissions data should be interpreted with the main caveat that we assume the relationships between underlying predictions our model has discovered for cities who report emissions data hold for these other cities. We acknowledge, however, that this is a major caveat to our results, but that the main goal of our study is to explore the potential strengths and limitations of an ML approach to developing a generalizable prediction model for city-level emissions that could be applied outside of Europe, given additional non-European city emissions data.

Despite these limitations, this research is a first step towards addressing the "lack of systematic knowledge on global contributions of cities to the Paris Agreement,"[25] which acknowledges the role of "all levels of government"[38] and seeks specific information regarding their impacts[39]. Few city actors participating in transnational climate initiatives report monitoring and inventory data, and even major cities claiming global climate leadership are absent from reporting[9,10,25,40]. Our study provides a consistent approach and time series data to investigate city-scale mitigation trends and performance, with potential for broadening the scope to areas outside of Europe.

Comparable and widespread emissions data are essential to support the Paris Agreement's "facilitative and catalytic"[41] mode and its "pledge and review and ratchet" mechanism designed to continuously evaluate national and subnational actors' progress and contributions to global mitigation efforts[42]. For virtuous, catalytic cycles supporting this process to occur, emissions data are needed to assess which actions are effective in driving mitigation and which entities are achieving reductions.

## Methods
### Dataset preparation
**Self-reported emissions inventory and climate action policy data.** Data for cities participating in the EUCoM were collected from two sources: Kona et al.[34], which provides a "verified and harmonized version" of the EUCoM data for 6200 member cities as of the end of 2019, and the EUCoM website itself. The Kona et al.[34] dataset for EUCoM cities includes self-reported emissions data (e.g., baseline or monitoring emissions inventories), as well as other characteristic data of the cities from the European Statistical Agency. We supplemented this dataset with more recent data for cities from the EUCoM website, which was scraped using the Beautiful Soup Python package[43] in February 2021. We primarily collected information on each cities' adhesion date to the EUCoM initiative, baseline emissions year, baseline emissions (in tons of carbon dioxide emissions or tCO$_2$), emissions reduction target, target year, and any reported inventory emissions (i.e., emissions data reported at a later year than a defined baseline year, from each city's Progress page). We also derived information regarding the cities' population and geographic coordinates (latitude/longitude) from the EUCoM website if available. Since Kona et al.[34] apply a series of statistical techniques to validate their dataset, we prioritized self-reported emissions data from this source if there were data available for a city both in Kona et al.[34] and the EUCoM website. Supplementary

Fig. 1 shows a scatterplot of the logged emissions data from both the EUCoM website and Kona et al.[34], which illustrates a strong correlation ($r^2 = 0.986$). In total, our dataset contained names of 7805 cities participating in the EUCoM initiative, with 6114 reporting any emissions information. We also imputed a 20% emissions reduction target by 2020 if no specific emissions reduction target was reported in Kona et al.[34] or on the EUCoM website for the purposes of the tracking progress analysis described in our previous study[10].

**Feature selection—predictors of urban climate emissions.** An important first step in building our predictive emissions model was determining a set of underlying predictors of city-level carbon emissions that would be universally available for all EUCoM cities and LAUs in Europe. We evaluated several predictors of urban greenhouse gas emissions to include as predictors in our model, based on existing literature regarding major sources and drivers of cities' emission profiles[6,44–46]. In terms of emission sources, the energy sector, specifically conversion of energy to electricity, is the largest source of urban greenhouse gas emissions, comprising around half upwards to 65% of total urban emissions, followed by the transportation sector (15–20%)[44]. Since stationary sources do not explain city greenhouse gas emissions in their entirety, we also investigated other proxies for major emissions sources, including heating and cooling demand, and air pollution variables such as fine particulate air pollution, which in cities results primarily from transport (~25%[47]), dust, and sulfur oxides (SO$_2$, SO$_4$). We also included population and gross domestic product (GDP) as relevant socioeconomic drivers of urban climate emissions[6], and evaluated a few country-level predictors, based on our previous study[10] that found national-level emissions reductions were predictors of city-level climate change performance, including country-level CO$_2$ emissions trend (2000–2018)[17] and carbon intensity of electricity-generation for the European Union[48].

Since high-resolution emissions data as a result of electricity production and consumption are not available for the vast majority of cities included in our analysis, we relied on the Open-Data Inventory for Anthropogenic Carbon Dioxide (ODIAC) database, which provides a globally gridded, annual 1 km × 1 km spatial resolution data of carbon dioxide emissions from fossil fuel combustion, cement production, and gas flaring from 2000 to 2019[49]. We selected the ODIAC dataset based on prior evaluation of its relevance for urban-level carbon emissions analysis, as described in Hsu et al.[10].

As proxies for building energy consumption due to heating and cooling, we downloaded monthly averaged, (0.5 × 0.625 degree or 55.5 × 69.375 km) spatial resolution land surface temperature data from the NASA MERRA-2 temperature product[50] and then calculated heating and cooling degree days (HDD and CDD, respectively) based on the number of monthly averaged measurements that deviate from a baseline temperature, $T_{base}$, which were then multiplied according to the number of days in each respective month (i.e., assuming the same HDD or CDD for each day of the month) and then summed across a year, according to the Eqs. (1–2) below:

$$HDD = \Sigma_m \left( T_{base} - T_i \right) \times Days_m{}^+ \tag{1}$$

$$CDD = \Sigma_m \left( T_i - T_{base} \right) \times Days_m{}^+ \tag{2}$$

where $T_{base} = 15.5$ degrees C for HDD and $T_{base} = 22$ degrees C for CDD[41] and $m$ is the month. For the EU model, we excluded cooling degree days since 99% of European cities had 0 cdd.

We included air pollution data extracted from satellite remote sensing resources. We included annual, gridded (~1 km) exposure to fine particulate matter pollution (PM$_{2.5}$) for years 2001–2020[51], since PM$_{2.5}$ pollution is generated from sources similar to carbon emissions in urban areas, mainly fossil fuel combustion from electricity

generation and transportation[52]. We also extracted several relevant air pollution variables from the MERRA-2 sensor, including dust surface mass concentration (DUSMASS), black carbon surface mass concentration (BCSMASS), sulfur dioxide surface mass concentration (SO2SMASS), and sulfate surface mass concentration (SO4SMASS).

We evaluated a few country-level predictors, based on a previous study[10] that found national-level emissions reductions were predictors of city-level climate change performance, including country-level $CO_2$ emissions trend (2000–2018)[17] and carbon intensity of electricity-generation for the European Union[53], although our final model did not include these variables, since they did not contribute significantly to the feature importance for our model (Fig. 1b).

We further accounted for population and gross domestic product (GDP) as relevant socioeconomic drivers of urban climate emissions[6]. For population, we used the Gridded Population of the World (GPW) dataset[54], which provides population estimates at a 1-km spatial resolution for five-year increments from 2000 to 2020. We calculated annual population estimates by linearly interpolating between these five-year increments. For GDP, we used a globally, annually gridded GDP per capita data at a 1-km spatial resolution from Kummu et al.[55], which provides data from 1990 to 2015. We spatially joined each LAU to its corresponding Nomenclature of Territorial Units for Statistics or NUTS (Level 3), for the European Union its International Territorial Unit, to derive a gross regional product (GRP) from the European Statistical Agency[56]. Since the NUTS3 GRP values are slightly broader in area than an LAU, we used the annual rate of change from 2016 to 2018 applied to the Kummu GDP data for each LAU to match the time series of the other spatial predictors.

**Spatially joining predictor variables with climate action participation dataset.** Since the original format of these predictor variables (e.g., fossil-fuel $CO_2$ emissions) are all gridded spatial data, we merged these datasets to each EUCoM city through spatial joins. We first collected the latitude and longitude of each city's centroid as provided by the various data sources. When the city centroids were not available from Kona et al.[34], EU Covenant of Mayors' website, or we determined errors in the geographic coordinates from either of these sources, we extracted the city centroids through Wikipedia's GeoHack website.

To determine each city's spatial boundaries, we used distinct approaches described below. For most of the cities, we collected data for local administrative units (LAUs), which are defined as "low-level administrative divisions of a country below that of a province, region or state," for all 28 European Union countries from the European Union's Statistical Agency[57]. The LAU data was spatially joined to our EUCoM city data frame in Python using the geopandas[58] package to associate each city with a LAU boundary for the purposes of matching additional predictor variables. We implemented a series of quality checks to ensure that the spatial joins were conducted correctly and to identify any issues in the geographic coordinates that may have been incorrectly specified on the EU Covenant website. These quality checks include (1) evaluating whether cities have the same geographic coordinates but are identified with distinct names; (2) comparing the reported population in the Kona et al.[34] or EUCoM website for an individual actor and the interpolated population after the spatial join; (3) examining any city with self-reported per capita emissions <0.2 tons per person or >40 tons per person; (4) compound annual growth rate in emissions is >−50% and <50%. These checks allowed us to determine whether there were any errors in the spatial join or underlying data collected for the EUCoM cities from either Kona et al.[34] or the EUCoM website.

Where manual corrections to LAUs also did not result in correct spatial joins, we utilized OpenStreet Map (OSM)[59] to get the correct boundary, particularly for large cities that may encompass more than one LAU. Supplementary Fig. 2 illustrates a few examples of the incorrect spatial join results and the fixed boundaries with OSM. After we verified the cities' boundaries, we then applied zonal statistics using the Python package rasterstats version 0.15.0[60], where each predictor variable was summarized for each city using its spatial boundary. Based on the definition of the predictor variables, we calculated mean values, except population, where we calculated the sum of all pixels that intersect with each city or LAU boundary.

**Model for predicting emissions and climate change performance.** Cities participating in the EUCoM are required to submit a Sustainable Energy (and Climate) Action Plan" (SE(C)AP) that includes a baseline emissions inventory, and a monitoring inventory every two years after that. Yet, at the time of data collection in February 2021, out of the nearly 10,000 signatories listed on the website, only 6114 actors had reported any emissions data, and only 1400 had reported more than one year of emissions monitoring data. We only included cities' data with an interpolated population greater than the 5th percentile (374 inhabitants) of the cities' population distribution. In total, 329 cities had populations below this threshold and were not included in the training or the prediction datasets. Consistent with Hsu et al. (2020), we also filtered out datapoints that reported <0.2 tons $CO_2$ per person or >40 tons $CO_2$ per person. The time period for self-reported emissions data ranged from 1990 to 2020, but we only used data >2000 (5880 unique actors with 6961 emissions datapoints) for the model training since this is the time period available for the predictor variables.

We further split our data into three subsets: the first subset used as training data includes all EUCoM cities that have at least one year of emissions data reported, whether its baseline emissions or a later inventory-year of data reported (EUCoM, 2021); a second subset are cities participating in the EUCoM but have not reported any emissions data; the third subset are cities not participating in the EUCoM. The first subset of reported emissions data to the EUCoM are used as training data to predict emissions for the latter two subsets of data. We applied the model built with the first dataset to these cities and predicted their likely emission of a given year. Supplementary Fig. 3 provides a flow diagram of the processing steps described above. Our training and test datasets were generated based on a standard 80/20 split of the data while preserving the underlying country representation (i.e., slightly over half of the available training data are from cities in Italy (52%), followed by Spain (26%)).

**Model selection - XGBoost**
We evaluated several regression models including multilinear regression, random forest, SVM, and extreme gradient boosting (XGBoost). The multilinear model is from the R base library; random forest and SVM are from R package caret version 6.0-86[61]; and XGBoost from XGBoost R package version 1.3.2.1[62]. We chose root mean square error (RMSE) and $r^2$ as the model comparison matrix to examine how each model performs on both the training and test datasets. For random forest, SVM, and XGBoost models that are controlled by a set of hyperparameters, we applied grid search with fivefold cross validation to the models to get the best parameters that result in the lowest RMSE. Supplementary Table 6 shows the hyperparameters we used in these three models. Missing values in independent variables are a common issue in ML-based models, and the models we evaluated handle missing values in different ways. The XGBoost model is capable of handling missing values without any imputation. Therefore, after we trained an XGboost model with complete data in all independent variables (referred as XGBoost-w/o NA), we also trained the XGBoost model with the data that may have NA values in the independent variables (referred as XGBoost-w/-NA in the following sections. Note that all NA values are dropped after we split the data into training and test sets, so that all train and test dataset are exactly the same for models besides XGBoost-w/ NA. Supplementary Table 6 shows the

train and test RMSE and $r^2$ of the best tuned models. Both the random forest and XGBoost model are tree-based regression models, and our results suggest that the tree based models perform better than other models for our dataset (Supplementary Table 6). In addition, the XGBoost-w/ NA model is trained with 357 more data points with NA values in the independent variables and achieved: RMSE = 155865.63 and $r^2$ = 0.90.

Based on the model training results and the capability of handling missing values, we decided to proceed with XGBoost. XGBoost stands for "extreme gradient boosting" and has gained popularity due to its high performance in machine-learning competitions such as Kaggle[63]. Gradient boosting models like XGBoost perform supervised regression tasks through an iterative approach to predict a target variable (i.e., emissions), optimizing predictive performance by combining multiple "weak" trees to fit new models that are more accurate predictors of a response variable[64,65]. One advantage of gradient-boosting machine learning models such as XGBoost is that they are robust to issues that are of concern in typical regression-based techniques, including multicollinearity issues[66,67]. A decision-tree consists of splits −iterative selections of features that separate data into two groups and then determine which is the optimal "split" on a feature based on the score. If two features or variables are correlated, then only one will be selected and the algorithm will not utilize information from the correlated feature since it has already been captured by the first. The XGBoost gradient-boosting model has been widely used in air quality monitoring[65,68,69] and greenhouse gas (GHG) emissions estimation[70] for its high efficiency, flexibility, and portability. Si and Du[65] further note additional advantages of XGBoost, which requires less data preprocessing and has fewer hyperparameters, parameters an ML model uses to control the learning process for tuning[71].

We utilized recursive feature elimination (RFE)[72], a machine learning technique that assists with feature selection to identify optimal features for a prediction or classification problem by eliminating the "weakest" features in a dataset[73]. Although RFE approaches may be more relevant for datasets that include several dozen or hundreds of variables, we implemented RFE using the FeatureTerminatoR[74] package in R, which suggested the inclusion of heating degree days, fossil-fuel CO2 emissions (odiac), fine particulate pollution (pm25), gdp per capita, population, population density, latitude, longitude, dust mass concentration, and emissions year. We evaluated alternate model specifications that included additional variables collected (e.g., sulfur dioxide emission concentrations), but their inclusion did not significantly improve the prediction accuracy of our model and we erred on model parsimony in our final model specification[75] (Supplementary Fig. 5 and Supplementary Table 7).

Our implementation of the XGBoost is determined by a set of hyperparameters, which are parameters the machine learning model uses to control the learning process[71]. These included the maximum depth of the tree, the learning rate, the minimum sum of weight in a node, minimum loss reduction, and the percent of rows to use in each tree which are the standard hyperparameters included in the XGBoost implementation in R[76]. To obtain the best hyperparameters set for the model and evaluate how the model performs, we first split our dataset with a 80/20 split sampling across countries, meaning we used 80% of the data as training data to predict the other 20% of the dataset[65]. We then conducted a grid search (Supplementary Table 4) on the hyperparameters with fivefold cross-validation to determine the model with the lowest mean root mean squared error. Supplementary Table 4 shows the hyperparameter ranges and the optimized values. Following the hyperparameter grid search, we trained the model with the training dataset with the best result from the hyperparameter grid search. We then tested the model accuracy using the test data.

The final model was built with the optimal parameter set from the grid search, which is the process of building models with all the possible parameter combinations and finding the best parameter set with

which the model performs the best on training samples. As Supplementary Table 4 describes, the optimum result for the model is achieved when max depth = 13, minimum child weight = 1, eta (learning rate) = 0.5, gamma = 1, and trains the model with 40 rounds, which achieved a mean absolute percentage error between the training and predicted values of 8%, and $r^2$ = 0.88 for the test data See Supplementary Fig. 4 for scatterplots of model performance. Supplementary Table 7 shows the results of a few selected alternate model specifications that were evaluated but ultimately not selected for predictions for other years and all other LAUs. Supplementary Fig. 4 shows scatter plots of the self-reported and predicted emissions for the training and test datasets. We used the XGBoost R package's built-in function *xgb.importance* to determine the final model's feature importance (i.e., which predictors have the greatest predictive or explanatory power)[76].

## Predicting 'likely' emissions levels for all entities 2001–2018

After building the final model with optimal parameters and evaluation, we applied our model to (1) EUCoM cities that do not report emissions (i.e., participating cities); and (2) all external LAUs in Europe that do not participate in the EUCoM. We bootstrapped 1000 predicted emissions intervals for each year for each actor to ensure robust median estimates. In addition to the optimum parameters from the grid search, we used the "subsample" parameter to introduce randomness into the model. This parameter determines the percent of rows in our dataset to use in each tree. We set this value to 0.90 and, so the model is built with 90% of the total dataset. We then calculated the 5th percentile, 95th percentile, mean, and median value for each predicted emissions estimates for each actor and year.

## Performance metrics

We calculated several performance metrics (e.g., linear trend in predicted emissions between 2001 and 2018, annual percentage change in emissions, and annualized percentage reduction in per capita emissions) using the predicted emissions data for each actor and evaluated them before utilizing the annualized percentage reduction in per capita emissions (annual per capita emissions trend) as our main evaluation metric, consistent with Hsu et al.[10], as described in Eq. 3.

$$reduction_c = -100 \times \frac{predemissions_{\min(year)} - predemissions_{\max(year)}}{predemissions_{\min(year)}}$$
$$\times \frac{1}{(year) - \min(year)}$$
(3)

Consistent with Hsu et al.[10], we determined whether a city is 'on track' to achieving their stated emission reduction goal or not, we calculated the ratio of actual (i.e., achieved) per capita emissions reduction in the inventory year to the targeted per capita emissions reduction in the inventory year, both in comparison to the baseline year, assuming that emissions reduction between the baseline year and the target year are pro-rated linearly (i.e., constant emissions reduction from one year to the next). More specifically, we define $\rho$ through the following Eqs. (4–7):

$$Reduction_{achieved} = Predemissions_{\min(year)} - Predemissions_{\max(year)}$$
(4)

where:

$Predemissions_{\min(year)}$ is predicted emissions per capita of the city in the minimum year for which predictor data are available. For most cities this was the year 2001;

$Predemissions_{\max(year)}$ is the predicted emissions per capita of the city in the maximum year for which predictor data are available. For

most cities this was the year 2018;

$$Timelapsed = (Year_{max} - Year_{min}) \div (Year_{target} - Year_{min}) \quad (5)$$

where:

$Year_{min}$ is the minimum year for which predicted emissions data are available

$Year_{max}$ is the maximum year for which predicted emissions are available

$Year_{target}$ is the year by which committed emissions reductions are to be achieved

$$Reduction_{required} = Predemissions_{min(year)} \times Target \times Timelapsed \quad (6)$$

where:

$Target$ is the committed emissions reduction of the city (percentage).

$$\rho = \frac{Reduction_{achieved}}{Reduction_{required}} \quad (7)$$

### Interrupted Time Series Analysis

To investigate whether participation in the EUCoM is associated with a change in a cities' emissions, we employed an interrupted time series (ITS) modeling approach[20] to compare trends in EUCoM cities' annual per capita emissions prior to and following their adhesion year. ITS designs evaluate an outcome for a population sample exposed to an intervention before and after, using repeated observations at regular intervals[19,77]. Although there is strong internal validity of an ITS design, there are limitations in terms of potential weak external validity in that the results may not be generalizable to other groups due to the fact that ITS cannot rule out the possibility of unmeasurable or uncontrolled factors leading to a change in the outcome variable.

We estimate annual percent changes in per capita emissions reductions ($pct.chg$) from 2001 to 2018 for each city ($i$) in country ($c$) for each year ($t$) with the following Eq. (8):

$$
\begin{aligned}
pct.chg_{i,c,t} = {} & \alpha_i + \beta_1 Time + \beta_2 Joined + \beta_3 TSJ + \beta_4 \log(GDP)_{i,c,t} \\
& + \beta_5 \log(pop\,density_{i,t,c}) + \beta_6 predicted\,emissions\,per\,capita_{i,t,c,} \\
& + \beta_7 emissions\,target_{i,t,c} + \beta_8 mitiation\,plan_{i,t,c} \\
& + \beta_9 mitigation\,inventory_{i,t,c} + \gamma_C + \delta_t + \epsilon_{i,c,t}
\end{aligned}
$$
$$\quad (8)$$

where $Time$ is a variable that indicates the number of years since a city adhered to the EUCoM initiative; $Joined$ is a dummy variable that indicates whether the observation refers to before (0) or after (1) the city adhered; $TSJ$ is the time elapsed since a city joined the EUCoM in years. We also control for differences between cities' population density, GDP per capita, emissions per capita predicted by our machine learning model, 2020 percentage reduction target, and whether the city has adopted a mitigation plan or conducted an emission inventory. We also include country dummies ($\gamma_C$) to control for unobserved, time-invariant factors common to cities within a country and year fixed effects ($\delta t$) to control for exogenous characteristics that may influence emissions in a given year.

### Software

Data scraping and geospatial data processing were conducted using python (version 3.68), Beautiful Soup package (version 4.8.2), geopandas (version 0.9.0), rasterio version (1.0.21), and rasterstats (version 0.15.0) and the R statistical programming environment (version 3.6.2). The machine learning model was developed and conducted in R using the XGBoost package (version 1.6.0.1)[76]. Figures were made using ggplot2[78] data

visualization package (version 3.3.6) and maps were made in QGIS (version 3.16).

### Reporting summary

Further information on research design is available in the Nature Portfolio Reporting Summary linked to this article.

### Data availability

The data generated in this study have been deposited in the Data-Driven EnviroLab Dataverse repository (https://doi.org/10.15139/S3/NRJ5ZO). Raw data collected, processed and utilized for this study include: the Open-Data Inventory for Anthropogenic Carbon Dioxide (ODIAC) database (https://doi.org/10.17595/20170411.001); NASA MERRA-2 monthly temperature product (https://doi.org/10.5067/KVIMOMCUO83U); NASA MERRA-2 monthly mean column mass density of aerosol components (black carbon, dust, sea salt, sulfate, and organic carbon), surface mass concentration of aerosol components (https://doi.org/10.5067/FH9A0MLJPC7N); Surface PM2.5 from the Atmospheric Composition and Analysis Group at Washington University at St. Louis (https://doi.org/10.1021/acs.est.1c05309); Gridded Population of the World dataset[54] (https://doi.org/10.7927/H4F47M2C) Globally gridded gross domestic product (GDP) data from Kummu et al.[55] (https://doi.org/10.1038/sdata.2018.4); Eurostat's Gross domestic product (GDP) at current market prices by NUTS 2 regions (http://data.europa.eu/88u/dataset/egT31kJF7IArVLXu1rTkQ); Kona et al.[34] Global Covenant of Mayors, a dataset of greenhouse gas emissions for 6200 cities in Europe and the Southern Mediterranean countries (https://doi.org/10.5194/essd-13-3551-2021); other data for the EU Covenant of Mayors cities were collected from (https://www.covenantofmayors.eu/); Local Administrative Units from the Eurostat database[56] (https://ec.europa.eu/eurostat/web/nuts/local-administrative-units); administrative boundaries of cities from OpenStreetMap (https://planet.openstreetmap.org); city centroids were extracted through Wikipedia's GeoHack website (https://www.mediawiki.org/wiki/GeoHack).

### Code availability

Code used for this study is available on the Data-Driven EnviroLab GitHub page (www.github.com/datadrivenenvirolab/citiesML) or upon reasonable request from the corresponding author.

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

## Acknowledgements

This research was supported by an IKEA Foundation Grant (Grant No. A19051; A.H.) and a 2018 National University of Singapore Early Career Research Award (Grant No. NUS_ECRA_FY18_P15; A.H.) awarded to A.H. We thank Zhi Yi Yeo and Vasu Namdeo of Yale-NUS College for assistance in data collection. We also thank Glenn Sheriff (Arizona State University), Joe Aldy (Harvard Kennedy School of Government), and Evan Johnson (University of North Carolina-Chapel Hill) for comments on an earlier version of this draft.

## Author contributions

A.H. conceived, co-designed study, collected data, conducted modeling and statistical analysis, made figures, and wrote the paper. X.W. collected data, conducted statistical modeling and validation, made figures, and contributed to the paper's writing. J.T. assisted with ML-model selection and implementation. W.T. assisted with data collection and merging. N.G. helped conceive and design the study, collect and process data, and interpret results.

## Competing interests

The authors declare no competing interests.
