## [Peer Review File · Nature Communications]

Predicting European cities' climate mitigation performance using machine learningREVIEWER COMMENTS

Reviewer #1 (Remarks to the Author):

This study responds a critical question of how to evaluate the mitigation performance for local and municipal actors in the context of less data availability. Undoubtedly, the application of machine learning methodology and open access emission inventory dataset is appropriate and could achieve above research objectives. As such, I believe this paper may arise widespread interest in the reader and to be suitable for publication in nature communications. However, in my review the quality of the paper could be further improved if the following issues were addressed.

(1) Because the key point of evaluating the mitigation performance for different actor is to predict accurately the amount of CO₂ emissions, if possible a bit more explained variables might be introduced into statistical model, such as statistical indicators of pollutions emissions (SO₂, dust), energy consumption/intensity, and relevant socioeconomic drivers. The biggest advantage to do this is that it may be further improved the accuracy of regression model, and linear/nonlinear relationships between urban CO₂ emissions and economic growth would be explored. Despite large scale sample size would increase the difficulty of data collection, the paper lacks enough expatiated variables constituting the statistical method as well as regarding the interpretation of results.

(2) In terms of urban classification used in current paper, is it necessary to distinguish the cities gorup achieving the reduction emission goals and none? Separate modeling between two types cities would be useful to explain the origin of mitigation performance differences.

(3) I think powerful feature selection technologies such as Recursive Feature Elimination and Boruta would be used to extract the important critical variables and input parameters for modeling CO₂ emissions for each city type group. This could avoid information redundancy generated by large number of features involved in the model.

(4) It would be better to provide more specified practical implications such as GHG emission reductions policies or actions adopted by each city type group. This would be more valuable for urban planning and future global sustainable development to fight with climate change.

(5) Is the XGBoost the best model compared with SVM, RF model? It often remains a bit unclear, which exact comparisons are made.

Reviewer #2 (Remarks to the Author):

After a very careful reading of the work entitled "Predicting European Cities' Climate Mitigation Performance using Machine Learning", I have found a very well-done work, well presented, and organized, clear in concepts and methodology. The manuscript is of interest to evaluate the mitigation performance based on the new developed machine learning methodology. The topic and context attract attention for many readers from various disciplines. The study is worth to be published after conducting the revisions.

(1) The study assessed the carbon emission performance of magnanimous European cities based on the data of nearly 50,000 local and municipal actors. For the study, mass data from different sources, including reports, statistics, site monitoring data and satellite data. This issue is what worries me most, i.e., the uncertainty of the model. As it should be, I have no doubt about the accuracy of these raw data. There is, however, significant differences for the statistical caliber and spatiotemporal resolution of these raw data. It will inevitably affect the accuracy of the mathematical algorithm. Therefore, it is necessary to discuss the uncertainty of the model and eliminate this seemingly unimportant factor that directly affects the final result.

(2) Fossil-fuel CO₂, population and emissions are well correlated. But here's an interesting question (Fig. 1). There is also a strong correlation between fossil-fuel CO₂ and population. This becomes an obvious collinearity problem in the model. According to the expression in the current results, it might as well study the population as a piecewise function, which may have better results.

Reviewer #3 (Remarks to the Author):

The manuscript presents a very interesting research that aims to shed light on emission reductions achieved by urban climate policies and strategies and to provide concrete evidence on the effect of cities' involvement in transnational climate initiatives in terms of GHG reductions. The focus is on European subnational governments and the Covenant of Mayors for Climate and Energy (EUCoM), which is one of the largest voluntary transnational climate initiatives.

The work demonstrates how to predict cities' emissions on an annual basis, which is very useful to estimate the performance of local and municipal actors and especially for partially overcoming the common scarcity of emissions data. The work is rooted in a robust integrated methodological approach that includes an innovative machine learning (ML)-based framework. This approach was developed and tested on a large-scale dataset of nearly "50,000 local and municipal actors" for which the authors had "underlying spatial data" "representing 301 million inhabitants" over a time horizon from 2001 to 2018.

The proposed approach is considered sound and appropriate to meet the research questions of the study. The pros and cons of the proposed approach are critically discussed, and the validity of the predictive model has been confirmed by the scatterplot of self-reported emissions in Figure 2 and also validated with other scientific studies.

The manuscript is well written and structured clearly and effectively. The abstract is clear and accessible. The introduction and conclusions are appropriate. Data interpretation and conclusions are very sound and robust. The findings and conclusions are very interesting to both the scientific community and decision makers at all levels of government, and the way they are presented is engaging, but it is not always easy to associate the data reported in the text with what is listed/presented in the tables and figures (both main and supplemental).

Therefore, some minor improvements would be appreciated to make the methodology and presentation of the results clearer, more transparent, and thus to facilitate its reproduction, as detailed below. In particular, it is requested to better clarify and make consistent the number of cities included in the analyzed sample (particularly in the three subgroups that compose it), in various parts of the manuscript (and related figures and tables).

The references are appropriate, but need to be double-checked to correct some inconsistencies, as detailed below.

Minor improvements suggested:

- Methods / "Dataset preparation" section

- o The section begins by saying that "Data for cities participating in the EUCoM were collected from two sources: " but then cites only one source, "Kona et al. (2021)". It is suggested that the reference to "two sources" be removed or that the second source, i.e., "the EUCoM website" be added as explained a few sentences later.

- Methods / "Self-reported emissions inventory and climate action policy data" section

- o It is said that "In total, our dataset contained names of 8,242 cities participating in the EUCoM initiative, with 6,309 reporting any emissions information". These number do not correspond to those reported both in Figure S6 (Participating EUCoM (n=1,906); Reporting EUCoM (n=5,628)) nor in the caption of Figure S6 where it is reported another different number for reporting cities ("Histograms comparing attributes of the study's three groups of cities 1) reporting (n=5,628); participating (n=1,936) and external cities (n=39,817)").

- o Then, the composition of the 3 groups of cities must be rechecked and made uniform throughout the main text, related figures, and supplementary materials.

- Results / "Trends in Performance" section:

- o In the following sentence, please clarify the meaning of "se:" in brackets: "... we find that joining the EUCoM is associated with a -1.64 (se: 0.13) percent annual per capita reduction"

- o In the following sentence "The EUCoM required cities to adopt at minimum a 20 percent reduction target by 2020 and at least a 40 percent reduction target by 2030..." it would be useful to clarify that this depends on when the plan was developed within the EUCoM: 2008-2015, SEAPs that commit to at least a 20% reduction in CO2 emissions by 2020: post-2015, SECAPs that commit to at least a 40% reduction in CO2 by 2030.

- o Regarding the definition of "ambitious"/"unambitious" provided for participating EUCoM cities, I

am somewhat uncertain whether a city can be defined as "ambitious" if, for example, it has set a GHG emission reduction target slightly above the 20% threshold (e.g., equal to 21%), considering that the time horizon analyzed extends to 2018 and, as we have already emphasized, from 2015 onwards the minimum target of EUCoM signatories is 40% by 2030 (SECAPs). Obviously, this is a simplification considering that there are many cities that have first defined a SEAP with a target of 20% by 2020 and have subsequently updated it to SECAP imposing a new target of 40% by 2030. Although the results of the study demonstrate that the threshold set for defining ambitious/unambitious cities do not affect the predicted model three classes of ambitiousness ("unambitious": less than 20%, "average ambitious": within 20% and 40%; "ambitious": more than 40%) would have better represented the current situation, also in light of the results of the study by Salvia et al. (2021) (<https://doi.org/10.1016/j.rser.2020.110253>). This study reveals, in fact, that 78% of the 327 cities in the sample (EU27+UK) have a local climate change plan with average mitigation targets just below 40% (specifically 39.45% for cities reporting CO₂ emissions or 39.57% for cities reporting CO₂eq).

o I would suggest adding the reference to Table 2 at the end of the following sentences "...reporting cities on average reduced per capita emissions 1.6 ± 2.0 from 2001 to 2018, while participating cities exhibited no or minimal reductions (-0.08 ± 1.5)."

o I would suggest adding the reference to Table 3 when you discuss the results obtained replicating the method used in Hsu et al. (2020), that is "Fifty-five percent of participating cities were on track to achieving their emissions reduction targets, with Scandinavian countries in the lead (87 percent in Denmark; 67 percent in Finland and Norway) ..."

o Some statements require additional information or references to tables/figures in order to facilitate understanding and verification of the data provided, as in the following cases:

- "While 84 percent of EUCoM cities have reduced emissions during this time period, only 35 percent of external cities achieved a negative trend in emissions reductions."

- "Within the EUCoM cities, we find that nearly 8,000 participating cities with 301 million inhabitants have reduced emissions 185.82 million tons between 2001-2018."

- "Thirty-eight percent of participating EUCoM cities achieved a greater annualized per capita emissions reduction after they joined the EUCoM, on average 3.67 ± 5.66 percent more than the year prior to their adhesion year."

• "Model for predicting emissions and climate change performance" section:

o I would suggest changing this sentence "Cities participating in the EUCoM are required to submit a Sustainable Energy Action Plan" to "Cities participating in the EUCoM are required to submit a Sustainable Energy (and Climate) Action Plan" (SE(C)AP).

o When referring to the EUCoM SEAP I would suggest skipping the word "monitoring" in "baseline emissions monitoring inventory" because the point of the BEI – Baseline Emissions Inventory is to provide a baseline against which the target is set while the monitoring report, at least every second year after submission of the action plan, allows to measure progress toward the targets set in the action plan.

o In this section, it is indeed discussed how starting from the initial list of "nearly 10,000 signatories listed on the website" with "only 6,114 actors" reporting any emissions data the authors after various calculations used "5,621 unique actors" "for the model training". This figure should match what is reported in the EUCoM Reporting (n=5,628) in Figure S6 or, if not, the difference between the two values justified.

o The same section then clarifies that the final sample of cities is composed of 3 subsets, of which appropriate definitions are provided and for each of which it would be useful to provide the same figures (number of cities) introduced in the Methods section and shown in Figure S6 and its caption.

• "Model selection - XGBoost" section:

o Please delete "has XGBoost" in the following sentence "The XGBoost gradient-boosting model has XGBoost has been widely used in air quality monitoring⁵⁷⁻⁵⁹"

• Tables

o Table 2 is cited in the text few lines before Table 1 (page 3).

o Table 2: in the second row of Table 2 it should be clarified if "EUCoM cities" means the sum of "reporting" and "participating" cities or only the reporting cities. Similarly, it should also be clarified whether "All LAUs" refers to all cities in the three sub-groups (reporting + participating +

external) or just the “external cities” as specified in the text (but in that a case it should be indicated as “All other LAUs”, as done in Table 1). If “All LAUs” refers to all cities then the wording “external cities” should be replaced with “All LAUs” to be consistent with what is stated on page 3: “Overall, we find that EUCoM cities have reduced emissions from 2001 to 2018 compared to external cities in the European Union that are not signatories (-1.22 ± 2.00 vs. 5.21 ± 11.03 annual per capita emissions trend; Table 2)”. It is suggested that a brief legend be added to clarify the wording used to label the different comparison groups.

o Table 4 is cited in the text few lines before Table 3 (page 4).

- Figures

o Figure 2 “Scatterplot of self-reported emissions”: It would be helpful to explain the use of the different colors as done at the bottom of Supplementary Figure 5 (although these colors can be deduced from Figure 2.b).

o There are two “Figures 3”: A) Predicted, self-reported emissions, and primary predictor variables for three cities of varying population sizes; and B) Annual per capita emissions reduction trend from 2001-2018 for cities with a population larger than 375 inhabitants (the 10th percentile of the cities included in the training data) participating in the EUCoM (left) and all other local administrative units (LAUs; right). If the authors intend to keep both, then the figures should be renumbered (becoming 6 instead of 5).

o Figure 3 “Predicted, self-reported emissions, and primary predictor variables for three cities of varying population sizes”: I would suggest making the red triangle indicating self-reported emissions more prominent because in London's case the 2013 value is almost completely hidden under the blue dot representing the predicted value.

o Figure S3 “Overview of methodological workflow and data processing steps” is much appreciated because it clarifies the connections and feedback between the different steps involved in the research. It would be helpful to add to this figure the number of cities included in each of the 3 subsamples represented by the rectangles labeled (a), (b), and (c).

- References:

o Some references are cited correctly in the text (without the number) but must be included in the reference list. This is the case, for example, of Marcotullio et al., 2013; Dodman, 2009; Rosa and Dietz, 2012; WRI CAIT, 2020; EUROSTAT, 2021; etc. in the section “Feature selection - Predictors of urban climate emissions”)

o Some references are correctly stated in the text in parentheses, but in some cases the reference number is skipped (e.g., Kona et al. (2016) in the Discussion);

o Reference numbers should also be included in Supplementary Table 1 (“Source” column), please note that one reference (Van Donkelaar et al (2020)) should be included in the list of references.

o It is suggested to check ref. #15 which seems to have some typos in the reference list “15. Moran, D. et al. Estimating CO₂ emissions for 108 000 European cities. Earth Syst. Sci. Data 14, 845–864 (2022).”

REVIEWER COMMENTS

Reviewer #1 (Remarks to the Author):

This study responds a critical question of how to evaluate the mitigation performance for local and municipal actors in the context of less data availability. Undoubtedly, the application of machine learning methodology and open access emission inventory dataset is appropriate and could achieve above research objectives. As such, I believe this paper may arise widespread interest in the reader and to be suitable for publication in nature communications. However, in my review the quality of the paper could be further improved if the following issues were addressed.

(1) Because the key point of evaluating the mitigation performance for different actor is to predict accurately the amount of CO₂ emissions, if possible a bit more explained variables might be introduced into statistical model, such as statistical indicators of pollutions emissions (SO₂, dust), energy consumption/intensity, and relevant socioeconomic drivers.

We thank the reviewer for this comment. To respond, we accessed additional data to consider and include in our modeling of European cities' emissions. The additional data we explored are detailed below as well as in the Methods section of the revised manuscript:

- CO₂ intensity of electricity generation by country data from the European Statistical Agency (<https://www.eea.europa.eu/ims/greenhouse-gas-emission-intensity-of-1>). Unfortunately these data were not available at the subnational or facility-level, and when we evaluated them for inclusion in our model, they did not rank highly on feature importance or gain.
- Number of coal and natural gas power plants - from the Global Energy Monitor. Unfortunately these data were quite sparse and resulted in 0 values for most of our entities. Including these data in our model worsened the predictive performance of our model, likely because we already capture large, stationary point-source emissions data from the ODIAC/fossil-fuel CO₂ emissions.
- Additional air pollution and dust data from MERRA-2 (https://disc.gsfc.nasa.gov/datasets/M2TMNXAER_5.12.4/summary?keywords=MERRA-2%20Aerosol%20Diagnostics), including:
 - BCSMASS = black carbon surface mass concentration
 - DMSSMASS = dimethyl sulfide surface mass concentration
 - DUSMASS = dust surface mass concentration
 - SO2SMASS = so₂ surface mass concentration
 - SO4SMASS = so₄ surface mass concentration
- We processed monthly measurements for these variables and then averaged on an annual basis for all subnational units at a spatial resolution of 0.5 ° x 0.625 ° (55km*70km).
- Relevant socioeconomic drivers - we already include population and GDP per capita, we also added population density.

We updated Figure 1's correlation coefficient matrix to show the relationship of these new air pollution and dust variables with EUCoM cities' self-reported emissions and the other variables we had previously used in the model.

The biggest advantage to do this is that it may be further improved the accuracy of regression model, and linear/nonlinear relationships between urban CO₂ emissions and economic growth would be explored. Despite large scale sample size would increase the difficulty of data collection, the paper lacks enough expatiated variables constituting the statistical method as well as regarding the interpretation of results.

We appreciate the reviewer's concern. We added more variables (see response above) to our model and conducted the recursive feature elimination (RFE) as the reviewer suggests below. Since many of the air pollution variables are correlated (see Figure 1) and have a low overall correlation with the cities' emissions, including Black Carbon, SO₂, SO₄ and Dust, the RFE selected only 1 of these variables (Dust) in addition to other variables we had already included in the model. We added Supplementary Figure 5, which shows that when all variables are included, only Dust contributes marginally to emissions prediction, and Black carbon, Dimethyl sulfide, SO₂, and SO₄ do not lend much by way of predictive power. We therefore only added Dust, since it contributed slightly more than the other air pollution variables, according to the Gain metric, which represents the fractional contribution of each feature to the model, with higher values indicating a more predictive feature. We implemented alternate model specifications that included additional variables, but when included in the model (see below Feature Importance: Gain), they did not add much by way of predictive power to the model and did not significantly improve the predictive accuracy of our model, likely given their overall low correlation with cities' self-reported emissions (see new Supplementary Table S5). We included some of these model results in the Supplementary Information Table S5 to demonstrate that the addition of more variables (e.g., SO₂, SO₄, dimethyl sulfide, black carbon) did not further improve the prediction or add much by way of predictive power (i.e., Gain) to the model. We further added more nuance to the interpretation of the results from the model in the Abstract, main results, discussion and limitations sections, to caution that our model is a first attempt to estimate likely emissions levels and mitigation performance at the urban scale, using limited relevant data that are consistently available for all urban administrative units in Europe.

Feature Importance: Gain

(2) In terms of urban classification used in current paper, is it necessary to distinguish the cities group achieving the reduction emission goals and none? Separate modeling between two types cities would be useful to explain the origin of mitigation performance differences.

We added a row in Table 2 to compare the performance (e.g., annualized per capita emissions reduction trends) of cities that are “on track” versus “not on track” (per Hsu et al., 2020, a pro-rated achievement measurement that assesses based on current rate of reduction whether a city will achieve or over-achieve its 2020 emissions reduction target). Since the cities’ emissions reduction targets are set for the year 2020 and the latest data we have available is for 2018, we cannot definitively compare whether cities have achieved their 2020 targets, only project a likely achievement (i.e., “on track”) based on the annualized per capita reductions achieved from 2001 to 2018. We agree with the reviewer that it would be interesting to explain the causal mechanisms or the origin of mitigation performance differences. We have instead, added a row to Table 2 that compares cities that are on track to achieving their 2020 mitigation target and those that are not on track. However, since we do not have information in this model regarding the actions cities take to achieve emissions reductions, it is not within the scope of this particular analysis to claim that we are able to distinguish or causally explain the mitigation performance differences between these groups of cities. We fleshed out these caveats in the Discussion section, starting with the paragraph, “While our study does not speak to causal mechanisms of the predicted emission trends...”

(3)I think powerful feature selection technologies such as Recursive Feature Elimination and Boruta would be used to extract the important critical variables and input parameters for modeling CO2 emissions for each city type group. This could avoid information redundancy generated by large number of features involved in the model.

We are grateful to the reviewer for making this suggestion. We implemented the reviewer’s suggestion and utilized the R Package (FeatureTerminatorR; Hutson, 2021) to conduct a recursive feature elimination. In addition to the variables we already included in our model, the RFE identified the inclusion of emissions_year, DUSMASS (dust mass concentration), and population density. We also tried alternate model specifications (Supplementary Figure S5 and Supplementary Table S5) and found that the RFE model yielded the best test r-squared for our test data and in our model evaluation.

We added the following paragraph to the Methods section: “We utilized recursive feature elimination (RFE) (Boughaci and Alkhawaldeh, 2021; Yan and Zhang, 2015), a machine learning technique that assists with feature selection to identify optimal features for a prediction or classification problem by eliminating the “weakest” features in a dataset (Ketu, 2022). We implemented RFE using the FeatureTerminatorR (Hutson, 2021) package in R, which suggested the inclusion of heating degree days, fossil-fuel CO2 emissions (odiac), fine particulate pollution (pm25), gdp per capita, population, population density, latitude, longitude, dust mass concentration, and emissions year. We evaluated alternate model specifications that included additional variables collected (e.g., sulfur dioxide emission concentrations), but their inclusion did not significantly improve the prediction accuracy of our model and we erred on model parsimony in our final model specification (Rasmussen and Ghararamani, 2000) (Supplementary Figure S5 and Supplementary Table S5).”

(4)It would be better to provide more specified practical implications such as GHG emission reductions policies or actions adopted by each city type group. This would be more valuable for urban planning and future global sustainable development to fight with climate change.

We thank the reviewer for this comment. We wholeheartedly agree that it would be valuable to include a discussion of the practical implications of policies and actions cities adopt to mitigate climate emissions. We have added a short exposition on these points to the Discussion section:

“Last, more research deeply evaluating the mitigation policies different groups of cities adopt to achieve emissions reductions and mitigation outcomes is required to inform urban planning and future climate policy development is needed.

(5)Is the XGBoost the best model compared with SVM, RF model? It often remains a bit unclear, which exact comparisons are made.

We thank the reviewer for this comment. We did compare XGBoost with SVM and Random Forest models and included the results in Supplementary Table 4. Based on the model training results and the

capability of handling missing values, we decided to proceed with XGBoost. We included more details regarding model selection in the Methods section:

“Based on the model training results and the capability of handling missing values, we decided to proceed with XGBoost. XGBoost stands for “extreme gradient boosting” and has gained popularity due to its high performance in machine-learning competitions such as Kaggle (Nielsen, 2016). Gradient boosting models like XGBoost perform supervised regression tasks through an iterative approach to predict a target variable (i.e., emissions), optimizing predictive performance by combining multiple “weak” trees to fit new models that are more accurate predictors of a response variable.^{55,56} One advantage of gradient-boosting machine learning models such as XGBoost is that they are robust to issues that are of concern in typical regression-based techniques, including multicollinearity issues (Alova et al., 2021; Hastie et al., 2009). A decision-tree consists of splits - iterative selections of features that separate data into two groups and then determine which is the optimal “split” on a feature based on the score. If two features or variables are correlated, then only one will be selected and the algorithm will not utilize information from the correlated feature since it has already been captured by the first. The gradient-boosting model has XGBoost has been widely used in air quality monitoring^{57–59} and greenhouse gas (GHG) emissions estimation⁶⁰ for its high efficiency, flexibility, and portability. Si and Du (2020)⁵⁶ further note additional advantages of XGBoost, which requires less data preprocessing and has fewer hyperparameters, parameters an ML model uses to control the learning process for tuning.⁶¹”

Supplementary Table 4. Training and Test results of multiple comparison models.

Model	Train_RMSE	Test_RMSE	Train_Rsquared	Test_Rsquared
XGBoost-w/ NA	24202.05	155865.63	0.9995	0.8999
Random Forest	197987.79	157619.88	0.9799	0.9005
XGBoost-w/o NA	24632.40	171360.38	0.9995	0.8799
SVM	524193.32	174429.23	0.8527	0.8923
Multilinear Regression	476764.12	200363.18	0.8302	0.8479

Reviewer #2 (Remarks to the Author):

After a very careful reading of the work entitled "Predicting European Cities' Climate Mitigation Performance using Machine Learning", I have found a very well-done work, well presented, and organized, clear in concepts and methodology. The manuscript is of interest to evaluate the mitigation performance based on the new developed machine learning methodology. The topic and context attract attention for many readers from various disciplines. The study is worth to be published after conducting the revisions.

(1) The study assessed the carbon emission performance of magnanimous European cities based on the data of nearly 50,000 local and municipal actors. For the study, mass data from different sources, including reports, statistics, site monitoring data and satellite data. This issue is what worries me most, i.e., the uncertainty of the model. As it should be, I have no doubt about the accuracy of these raw data. There is, however, significant differences for the statistical caliber and spatiotemporal resolution of these raw data. It will inevitably affect the accuracy of the mathematical algorithm. Therefore, it is necessary to discuss the uncertainty of the model and eliminate this seemingly unimportant factor that directly affects the final result.

We appreciate the reviewer's concern and wholeheartedly agree that there are certainly limitations and uncertainty regarding our approach. We have expanded further on the uncertainties inherent in the model in the Limitations Section and moved these discussions more prominently in the Discussion section to make clear the uncertainties involved in our model and their implications. In addition to the areas of uncertainty we already identified, mainly uncertainties in the self-reported emissions inventory data being used as training data for the wider group of European cities, we have included another paragraph in this section to speak specifically to the limitations of a machine learning-based approach for emissions prediction.

"Last, there are limitations to machine learning-based approaches for prediction, which have been identified and classified by (Kapoor and Narayanan, 2022). Since machine learning approaches are inherently stochastic (Sabuncu, 2020), the introduction of randomness to enhance model generalizability, which is seen as an advantage of ML approaches compared to traditional gaussian regression methods, risks a model's potential reproducibility (Kapoor and Narayanan, 2022). Our estimates, therefore only represent a likely bootstrapped median emissions level using the specific parameters tuned on the training subsample and set at a particular seed or initialization by the computing environment. In particular, our predictions of LAUs that report no emissions data should be interpreted with the main caveat that we assume the relationships between underlying predictions our model has discovered for cities who report emissions data hold for these other cities. We acknowledge, however, that this is a major caveat to our results, but that the main goal of our study is to explore the potential strengths and limitations of an ML approach to developing a generalizable prediction model for city-level emissions that could be applied outside of Europe, given additional non-European city emissions data."

(2) Fossil-fuel CO₂, population and emissions are well correlated. But here's an interesting question (Fig. 1). There is also a strong correlation between fossil-fuel CO₂ and population. This becomes an

obvious collinearity problem in the model. According to the expression in the current results, it might as well study the population as a piecewise function, which may have better results.

We thank the reviewer for this comment. One advantage of gradient-boosting machine learning models like the one we used (XGBoost) is that they are robust to issues that are of concern in typical regression-based techniques, including multicollinearity issues (Alova et al., 2021; Hastie et al., 2009). A decision-tree consists of splits - iterative selections of features that separate data into two groups and then determine which is the optimal "split" on a feature based on the score. If two features or variables are correlated, then only one will be selected and the algorithm will not utilize information from the correlated feature since it has already been captured by the first. We have included these additional references and this explanation in the Methods section:

"One advantage of gradient-boosting machine learning models such as XGBoost is that they are robust to issues that are of concern in typical regression-based techniques, including multicollinearity issues (Alova et al., 2021; Hastie et al., 2009). A decision-tree consists of splits - iterative selections of features that separate data into two groups and then determine which is the optimal "split" on a feature based on the score. If two features or variables are correlated, then only one will be selected and the algorithm will not utilize information from the correlated feature since it has already been captured by the first."

Reviewer #3 (Remarks to the Author):

The manuscript presents a very interesting research that aims to shed light on emission reductions achieved by urban climate policies and strategies and to provide concrete evidence on the effect of cities' involvement in transnational climate initiatives in terms of GHG reductions. The focus is on European subnational governments and the Covenant of Mayors for Climate and Energy (EUCoM), which is one of the largest voluntary transnational climate initiatives.

The work demonstrates how to predict cities' emissions on an annual basis, which is very useful to estimate the performance of local and municipal actors and especially for partially overcoming the common scarcity of emissions data. The work is rooted in a robust integrated methodological approach that includes an innovative machine learning (ML)-based framework. This approach was developed and tested on a large-scale dataset of nearly "50,000 local and municipal actors" for which the authors had "underlying spatial data" "representing 301 million inhabitants" over a time horizon from 2001 to 2018. The proposed approach is considered sound and appropriate to meet the research questions of the study. The pros and cons of the proposed approach are critically discussed, and the validity of the predictive model has been confirmed by the scatterplot of self-reported emissions in Figure 2 and also validated with other scientific studies.

The manuscript is well written and structured clearly and effectively. The abstract is clear and accessible. The introduction and conclusions are appropriate. Data interpretation and conclusions are very sound and robust. The findings and conclusions are very interesting to both the scientific community and decision makers at all levels of government, and the way they are presented is engaging, but it is not

always easy to associate the data reported in the text with what is listed/presented in the tables and figures (both main and supplemental).

Therefore, some minor improvements would be appreciated to make the methodology and presentation of the results clearer, more transparent, and thus to facilitate its reproduction, as detailed below. In particular, it is requested to better clarify and make consistent the number of cities included in the analyzed sample (particularly in the three subgroups that compose it), in various parts of the manuscript (and related figures and tables).

The references are appropriate, but need to be double-checked to correct some inconsistencies, as detailed below.

We really appreciate the reviewer's kind and supportive comments here.

Minor improvements suggested:

- Methods / "Dataset preparation" section

- o The section begins by saying that "Data for cities participating in the EUCoM were collected from two sources: " but then cites only one source, "Kona et al. (2021)". It is suggested that the reference to "two sources" be removed or that the second source, i.e., "the EUCoM website" be added as explained a few sentences later.

Thank you for spotting this typo. We changed the first sentence to: "Data for cities participating in the EUCoM were collected from two sources: Kona et al. (2021), which provides a "verified and harmonized version" of the EUCoM data for 6,200 member cities as of the end of 2019, and the EUCoM website itself."

- Methods / "Self-reported emissions inventory and climate action policy data" section

- o It is said that "In total, our dataset contained names of 8,242 cities participating in the EUCoM initiative, with 6,309 reporting any emissions information". These number do not correspond to those reported both in Figure S6 (Participating EUCoM (n=1,906); Reporting EUCoM (n=5,628)) nor in the caption of Figure S6 where it is reported another different number for reporting cities ("Histograms comparing attributes of the study's three groups of cities 1) reporting (n=5,628); participating (n=1,936) and external cities (n=39,817)").

- o Then, the composition of the 3 groups of cities must be rechecked and made uniform throughout the main text, related figures, and supplementary materials.

We thank the reviewer for this comment and have made all of the numbers of cities in each group consistent.

- Results / "Trends in Performance" section:

- o In the following sentence, please clarify the meaning of "se:" in brackets: "... we find that joining the EUCoM is associated with a -1.64 (se: 0.13) percent annual per capita reduction"

"se" is "standard error". We have made this clarification in the text.

- o In the following sentence "The EUCoM required cities to adopt at minimum a 20 percent reduction target by 2020 and at least a 40 percent reduction target by 2030..." it would be useful to clarify that this

depends on when the plan was developed within the EUCoM: 2008-2015, SEAPs that commit to at least a 20% reduction in CO2 emissions by 2020; post-2015, SECAPs that commit to at least a 40% reduction in CO2 by 2030.

We modified that sentence as follows: “Prior to 2015, the EUCoM required cities to adopt at minimum a 20 percent reduction target by 2020; post 2015, cities have been required to commit to and at least a 40 percent reduction target by 2030.”

o Regarding the definition of "ambitious"/"unambitious" provided for participating EUCoM cities, I am somewhat uncertain whether a city can be defined as "ambitious" if, for example, it has set a GHG emission reduction target slightly above the 20% threshold (e.g., equal to 21%), considering that the time horizon analyzed extends to 2018 and, as we have already emphasized, from 2015 onwards the minimum target of EUCoM signatories is 40% by 2030 (SECAPs). Obviously, this is a simplification considering that there are many cities that have first defined a SEAP with a target of 20% by 2020 and have subsequently updated it to SECAP imposing a new target of 40% by 2030. Although the results of the study demonstrate that the threshold set for defining ambitious/unambitious cities do not affect the predicted model three classes of ambitiousness ("unambitious": less than 20%, "average ambitious": within 20% and 40%; "ambitious": more than 40%) would have better represented the current situation, also in light of the results of the study by Salvia et al. (2021) (<https://doi.org/10.1016/j.rser.2020.110253>). This study reveals, in fact, that 78% of the 327 cities in the sample (EU27+UK) have a local climate change plan with average mitigation targets just below 40% (specifically 39.45% for cities reporting CO2 emissions or 39.57% for cities reporting CO2eq).

o I would suggest adding the reference to Table 2 at the end of the following sentences “...reporting cities on average reduced per capita emissions 1.6 ± 2.0 from 2001 to 2018, while participating cities exhibited no or minimal reductions (-0.08 ± 1.5).”

Done.

o I would suggest adding the reference to Table 3 when you discuss the results obtained replicating the method used in Hsu et al. (2020), that is “Fifty-five percent of participating cities were on track to achieving their emissions reduction targets, with Scandinavian countries in the lead (87 percent in Denmark; 67 percent in Finland and Norway) ...”

Done.

o Some statements require additional information or references to tables/figures in order to facilitate understanding and verification of the data provided, as in the following cases:

- “While 84 percent of EUCoM cities have reduced emissions during this time period, only 35 percent of external cities achieved a negative trend in emissions reductions.”

- “Within the EUCoM cities, we find that nearly 8,000 participating cities with 301 million inhabitants have reduced emissions 185.82 million tons between 2001-2018.”

- “Thirty-eight percent of participating EUCoM cities achieved a greater annualized per capita emissions reduction after they joined the EUCoM, on average 3.67 ± 5.66 percent more than the year prior to their adhesion year. “

We have modified and fixed these sentences.

- “Model for predicting emissions and climate change performance” section:

- o I would suggest changing this sentence “Cities participating in the EUCoM are required to submit a Sustainable Energy Action Plan” to “Cities participating in the EUCoM are required to submit a Sustainable Energy (and Climate) Action Plan” (SE(C)AP).

Done.

- o When referring to the EUCoM SEAP I would suggest skipping the word “monitoring” in “baseline emissions monitoring inventory” because the point of the BEI – Baseline Emissions Inventory is to provide a baseline against which the target is set while the monitoring report, at least every second year after submission of the action plan, allows to measure progress toward the targets set in the action plan.

Removed the word “monitoring” for that sentence.

- o In this section, it is indeed discussed how starting from the initial list of “nearly 10,000 signatories listed on the website” with “only 6,114 actors” reporting any emissions data the authors after various calculations used “5,621 unique actors” “for the model training”. This figure should match what is reported in the EUCoM Reporting (n=5,628) in Figure S6 or, if not, the difference between the two values justified.

Fixed. To address other reviewer’s comments, we added in some new variables, which affected the number of cities we could use for model training, so the numbers have changed slightly. We have triple checked to ensure that the number of cities for each figure and analysis step is made clear.

- o The same section then clarifies that the final sample of cities is composed of 3 subsets, of which appropriate definitions are provided and for each of which it would be useful to provide the same figures (number of cities) introduced in the Methods section and shown in Figure S6 and its caption.

Fixed.

- “Model selection - XGBoost” section:

- o Please delete “has XGBoost” in the following sentence “The XGBoost gradient-boosting model has XGBoost has been widely used in air quality monitoring^{57–59}”

Done

- Tables

- o Table 2 is cited in the text few lines before Table 1 (page 3).

Fixed.

- o Table 2: in the second row of Table 2 it should be clarified if “EUCoM cities” means the sum of “reporting” and “participating” cities or only the reporting cities. Similarly, it should also be clarified whether “All LAUs” refers to all cities in the three sub-groups (reporting + participating + external) or just the “external cities” as specified in the text (but in that a case it should be indicated as “All other LAUs”, as done in Table 1). If “All LAUs” refers to all cities then the wording “external cities” should be replaced with “All LAUs” to be consistent with what is stated on page 3: “Overall, we find that EUCoM cities have reduced emissions from 2001 to 2018 compared to external cities in the European Union that

are not signatories (-1.22 ± 2.00 vs. 5.21 ± 11.03 annual per capita emissions trend; Table 2)". It is suggested that a brief legend be added to clarify the wording used to label the different comparison groups.

Thank you for this comment. We made clarifications and standardized wordings when referencing different groups throughout the text and added a caption to Table 2 to clarify that 'EUCoM cities' include both reporting and participating cities.

o Table 4 is cited in the text few lines before Table 3 (page 4).

Fixed.

- Figures

o Figure 2 "Scatterplot of self-reported emissions": It would be helpful to explain the use of the different colors as done at the bottom of Supplementary Figure 5 (although these colors can be deduced from Figure 2.b).

Caption added to this figure: "Colors represent different countries, which are expanded in panel b."

o There are two "Figures 3": A) Predicted, self-reported emissions, and primary predictor variables for three cities of varying population sizes; and B) Annual per capita emissions reduction trend from 2001-2018 for cities with a population larger than 375 inhabitants (the 10th percentile of the cities included in the training data) participating in the EUCoM (left) and all other local administrative units (LAUs; right). If the authors intend to keep both, then the figures should be renumbered (becoming 6 instead of 5).

Fixed

o Figure 3 "Predicted, self-reported emissions, and primary predictor variables for three cities of varying population sizes": I would suggest making the red triangle indicating self-reported emissions more prominent because in London's case the 2013 value is almost completely hidden under the blue dot representing the predicted value.

Fixed

o Figure S3 "Overview of methodological workflow and data processing steps" is much appreciated because it clarifies the connections and feedback between the different steps involved in the research. It would be helpful to add to this figure the number of cities included in each of the 3 subsamples represented by the rectangles labeled (a), (b), and (c).

Fixed.

- References:

o Some references are cited correctly in the text (without the number) but must be included in the reference list. This is the case, for example, of Marcotullio et al., 2013; Dodman, 2009; Rosa and Dietz, 2012; WRI CAIT, 2020; EUROSTAT, 2021; etc. in the section "Feature selection - Predictors of urban climate emissions")

Fixed.

o Some references are correctly stated in the text in parentheses, but in some cases the reference number is skipped (e.g., Kona et al. (2016) in the Discussion);

Fixed.

o Reference numbers should also be included in Supplementary Table 1 ("Source" column), please note that one reference (Van Donkelaar et al (2020)) should be included in the list of references.

Fixed.

o It is suggested to check ref. #15 which seems to have some typos in the reference list “ 15. Moran, D. et al. Estimating CO₂ emissions for 108 000 European cities. Earth Syst. Sci. Data 14, 845–864 (2022).”

Fixed.

REVIEWERS' COMMENTS

Reviewer #1 (Remarks to the Author):

The current manuscript is good enough. Congratulations for authors! I have no other suggestions for this paper.

Reviewer #2 (Remarks to the Author):

This manuscript had been revised according to the comments of the reviewers. I have no further comments. The current version of the manuscript is worth publishing.

Reviewer #3 (Remarks to the Author):

The manuscript has been substantially improved in different aspects: greater clarity about the data used and in the presentation of the results, greater robustness in the description of the methods and tools adopted, honest discussion of the limitations of the research.

Therefore, it is considered that the authors have fully satisfied the suggestions made by the Reviewers in the previous review.

Thank you for your efforts

REVIEWERS' COMMENTS

Response to all reviewers: We greatly appreciate your efforts reviewing our paper and substantially improving it. Thank you!

Reviewer #1 (Remarks to the Author):

The current manuscript is good enough. Congratulations for authors! I have no other suggestions for this paper.

Reviewer #2 (Remarks to the Author):

This manuscript had been revised according to the comments of the reviewers. I have no further comments. The current version of the manuscript is worth publishing.

Reviewer #3 (Remarks to the Author):

The manuscript has been substantially improved in different aspects: greater clarity about the data used and in the presentation of the results, greater robustness in the description of the methods and tools adopted, honest discussion of the limitations of the research.

Therefore, it is considered that the authors have fully satisfied the suggestions made by the Reviewers in the previous review.

Thank you for your efforts